# Deciphering microbial interactions in synthetic human gut microbiome communities

Ophelia S Venturelli[1,*] , Alex C Carr[2,†], Garth Fisher[2,†], Ryan H Hsu[3] , Rebecca Lau[2],
Benjamin P Bowen[2], Susan Hromada[1], Trent Northen[2] & Adam P Arkin[2,3,4,5]

## Abstract

The ecological forces that govern the assembly and stability of the human gut microbiota remain unresolved. We developed a generalizable model-guided framework to predict higher-dimensional consortia from time-resolved measurements of lower-order assemblages. This method was employed to decipher microbial interactions in a diverse human gut microbiome synthetic community. We show that pairwise interactions are major drivers of multi-species community dynamics, as opposed to higher-order interactions. The inferred ecological network exhibits a high proportion of negative and frequent positive interactions. Ecological drivers and responsive recipient species were discovered in the network. Our model demonstrated that a prevalent positive and negative interaction topology enables robust coexistence by implementing a negative feedback loop that balances disparities in monospecies fitness levels. We show that negative interactions could generate history-dependent responses of initial species proportions that frequently do not originate from bistability. Measurements of extracellular metabolites illuminated the metabolic capabilities of monospecies and potential molecular basis of microbial interactions. In sum, these methods defined the ecological roles of major human-associated intestinal species and illuminated design principles of microbial communities.

**Keywords** ecology; human gut microbiome; mathematical modeling; microbial community; microbial interaction
**Subject Categories** Microbiology, Virology & Host Pathogen Interaction; Quantitative Biology & Dynamical Systems; Synthetic Biology & Biotechnology
**Mol Syst Biol. (2018) 14: e8157**

See also: **C Abreu _et al_** (June 2018)

## Introduction

Microbes have evolved in diverse microbial communities that occupy nearly every environment on Earth, spanning extreme environments such as acid mine drains and hot springs to multicellular organisms. The gut microbiome is a dense collection of microorganisms that inhabits the human gastrointestinal tract (Lozupone _et al_, 2012; Earle _et al_, 2015; Tropini _et al_, 2017) and performs numerous functions to impact human physiology, nutrition, behavior, and development (Ley _et al_, 2005; Fischbach & Sonnenburg, 2011; Foster & McVey Neufeld, 2013; Louis _et al_, 2014; Sharon _et al_, 2014; Rooks & Garrett, 2016). Functions of the gut microbiota are partitioned among genetically distinct populations that interact to perform complex chemical transformations and exhibit emergent properties such as colonization resistance at the community level. Such collective functions are realized by the combined interactions of diverse microbial species operating on multiple time and spatial scales and could not be achieved by a single monospecies population. The degree of spatial structuring in the gut microbiota varies across length scales: At a macroscale of hundreds of micrometers, bacteria cluster into distinct habitats, whereas at a scale of micrometers, intermixing of community members has been observed (Donaldson _et al_, 2015; Earle _et al_, 2015; Mark Welch _et al_, 2017).

The gut microbiota is composed of hundreds of bacterial species, the majority of which span the _Firmicutes_, _Bacteroidetes,_ and _Actinobacteria_ phyla (Ley _et al_, 2006). Constituent strains of the gut microbiota have been shown to persist in an individual over long periods of time, demonstrating that the gut microbiota exhibits stability over time (Faith _et al_, 2013). Perturbations to the system such as dietary shifts or antibiotic administration can shift the operating point of the gut microbiota to an alternative state (Relman, 2012). While the identities of the organisms and microbial co-occurrence relationships across individuals have been elucidated (Faust _et al_, 2012), we lack a quantitative understanding of how microbial interactions shape community assembly, stability, and response to perturbations. For example, the ecological and molecular forces that enable stable coexistence of the dominant phyla _Firmicutes_,

1 Department of Biochemistry, University of Wisconsin-Madison, Madison, WI, USA
2 Environmental Genomics and Systems Biology, Lawrence Berkeley National Laboratory, Berkeley, CA, USA
3 California Institute for Quantitative Biosciences, University of California Berkeley, Berkeley, CA, USA
4 Department of Bioengineering, University of California Berkeley, Berkeley, CA, USA
5 Energy Biosciences Institute, University of California Berkeley, Berkeley, CA, USA
*Corresponding author. Tel: +1 608 263 7017; E-mail: venturelli@wisc.edu
†These authors contributed equally to this work

*Bacteroidetes* and *Actinobacteria* are not well understood (Fischbach & Sonnenburg, 2011).

The resilience of microbiomes, defined as the capacity to recover from perturbations, is strongly linked to microbial diversity. Indeed, a reduction in microbial diversity of the human gut microbiome is associated with multiple diseases, suggesting that a high-dimensional and functionally heterogeneous ecosystem promotes human health (Sommer *et al*, 2017). Understanding the molecular and ecological factors influencing the stability and resilience of the gut microbiota has implications for the development of targeted interventions to modulate microbiome states. Central to this problem is inferring unknown microbial interactions and developing tools to predict temporal changes in community behaviors in response to environmental stimuli.

Cooperation and competition generate positive and negative feedbacks in microbial communities and influence functional activities and stability. Negative interactions have been shown to dominate microbial inter-relationships in synthetic aquatic microcosms (Foster & Bell, 2012). However, the prevalence of competition and cooperation in microbial communities occupying other diverse environments such as the human gut microbiota remains elusive. Direct negative interactions in microbial consortia can originate from competition for resources or space, biomolecular warfare, or production of toxic waste products (Hibbing *et al*, 2010). Positive interactions can stem from secreted metabolites that are utilized by a community member or detoxification of the environment. Pairwise microbial interactions can be modified by a third organism, leading to higher-order effects that influence community behaviors (Bairey *et al*, 2016). Ecological driver species, which exhibit a large impact on community structure and function, represent key nodes in the network that could be manipulated to control community states (Gibson *et al*, 2016).

Predicting community dynamics is a key step toward understanding the organizational principles of microbial communities. Computational models at different resolutions can be used to analyze and predict the behaviors of microbial communities (Faust & Raes, 2012). Dynamic computational models can be used to investigate temporal changes in community structure, and tools from dynamical systems theory can be used to analyze system properties including stability and parameter sensitivity (Astrom & Murray, 2010). Generalized Lotka–Volterra (gLV) is an ordinary differential equation model that represents microbial communities with a limited number of parameters that can be deduced from time-series data.

Here, we develop a systematic modeling and experimental pipeline to construct a predictive computational model of microbial community dynamics and interrogate microbial interactions mediating community assembly. Time-resolved measurements of monospecies and pairwise assemblages were used to train a dynamic computational model of a diverse synthetic human gut microbiome community. Our model revealed a high proportion of negative and frequent positive interactions. Specific ecological driver species and responsive organisms to community context were identified in the network. *Bacteroidetes* exhibited an overall negative impact on the community, whereas specific members of *Actinobacteria* and *Firmicutes* displayed numerous positive outgoing interactions. A prevalent pairwise sub-network composed of positive and negative interactions exhibited robust species coexistence to variations in model parameters. Our model showed that the majority of history-dependent responses in pairwise consortia were due to slow convergence to a steady state composition and these networks were enriched for negative interactions. The metabolic capabilities of monospecies were elucidated using exo-metabolomics profiling, and these data pinpointed a set of metabolites predicted to mediate positive and negative interactions. However, the metabolite profiles failed to forecast specific influential organisms modulating community assembly. Together, these results show that combinations of pairwise interactions can represent the assembly of multi-species communities and such pairwise couplings can realize a diverse repertoire of dynamic behaviors.

# Results

## Probing the temporal behaviors of monospecies and pairwise assemblages

We aimed to dissect the microbial interactions influencing community assembly in a reduced complexity model gut community spanning the major phyla *Bacteroidetes*, *Firmicutes*, *Actinobacteria,* and *Proteobacteria*. To this end, a synthetic ecology encompassing prevalent human-associated intestinal species *Bacteroides thetaiotaomicron* (BT), *Bacteroides ovatus* (BO), *Bacteroides uniformis* (BU), *Bacteroides vulgatus* (BV), *Blautia hydrogenotrophica* (BH), *Collinsella aerofaciens* (CA), *Clostridium hiranonis* (CH), *Desulfovibrio piger* (DP), *Eggerthella lenta* (EL), *Eubacterium rectale* (ER), *Faecalibacterium prausnitzii* (FP), and *Prevotella copri* (PC) was designed to mirror the functional and phylogenetic diversity of the natural system (Fig 1A; Qin *et al*, 2010). These species have been shown to contribute significantly to human health and are implicated in multiple human diseases (Watterlot *et al*, 2008; Larsen *et al*, 2010; Thota *et al*, 2011; Fujimoto *et al*, 2013; Haiser *et al*, 2013; Scher *et al*, 2013; Table 1).

Synthetic assemblages were arrayed in microtiter plates in an anaerobic chamber using an automated liquid-handling procedure (see Materials and Methods). A rich media (see Materials and Methods) was selected to support the growth of all monospecies. The communities were serially transferred at 24-h intervals to prevent strains with long lag phases from being eliminated and allow communities to approach a steady state composition by monitoring assembly over many cell generations. Further, serial transfers can also reflect recurrent temporal perturbations to the gut microbiota such as diet and colonic transit time (Fig 1B). Multiplexed 16S rRNA gene sequencing was performed in approximately 12-h intervals to elucidate the temporal variations in community structure at different community growth stages. The relative abundance for each species was computed as the sum of the read counts for each organism divided by the total number of reads per condition (see Materials and Methods). Since model construction is aided by absolute abundance information (Bucci *et al*, 2016; Widder *et al*, 2016), the total biomass of the communities was monitored approximately every 30 min using absorbance at 600 nm (OD600). Cellular traits such as cell adhesion, size, and shape can influence OD600 measurements (Stevenson *et al*, 2016). In addition, counting of colony-forming units (CFU) is biased by cell adhesion, dormant sub-populations, growth selection on solid vs. liquid media, and growth stage (Jansson & Prosser, 1997; Volkmer & Heinemann, 2011; Ou *et al*,

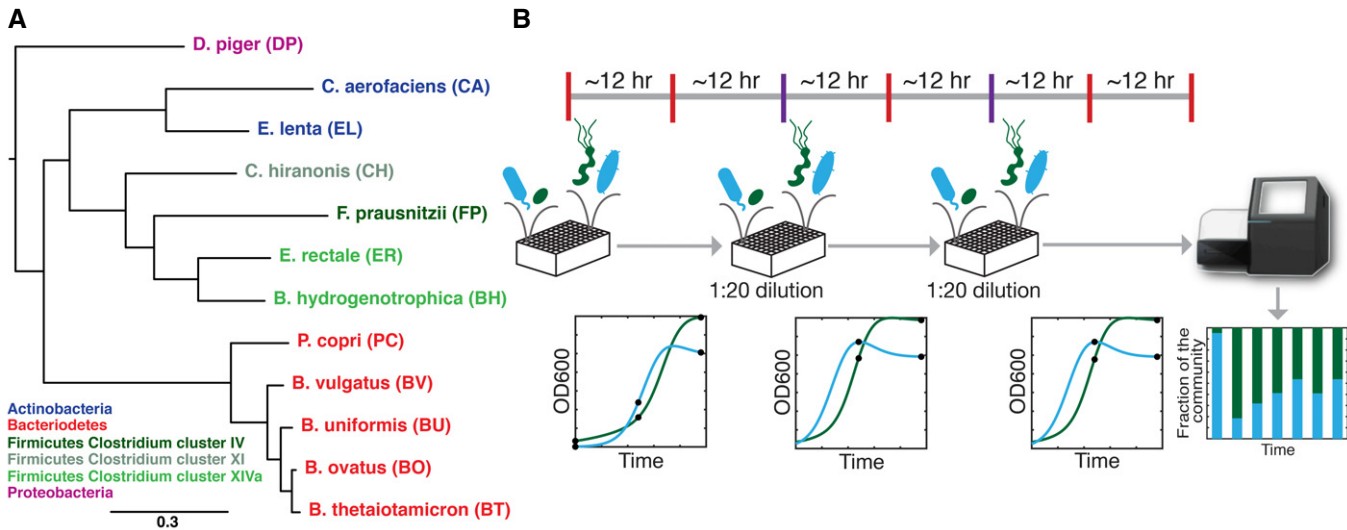

**Figure 1.** **Experimental design for high-throughput characterization of synthetic human gut microbiome consortia.**

A  Phylogenetic tree of the 12-member synthetic ecology spanning the major phyla in the gut microbiome including *Actinobacteria*, *Bacteroidetes*, *Firmicutes*, and *Proteobacteria*. Phylogenetic analysis was performed using a concatenated alignment of single-copy marker genes obtained via PhyloSift (preprint: Darling *et al*, 2014). Maximum likelihood trees were generated using default options. The scale bar represents the number of substitutions per site in the alignment.

B  Schematic of the experimental design for this study. Species were combined using an approximately 1:1 or 19:1 initial proportion based on absorbance measurements at 600 nm (OD600) into microtiter plates using liquid-handling robotic manipulation. Approximately every 12 h, samples were collected for multiplexed 16S rRNA gene sequencing (black circles). Relative abundance was measured using multiplexed 16S rRNA gene sequencing of the V3–V4 region using dual-indexed primers compatible with an Illumina platform (stacked bar plot, bottom right). Serial transfers were performed in approximately 24 intervals (purple bars, top) by transferring an aliquot of the communities into fresh media using a 1:20 dilution. In parallel, time-resolved OD600 measurements of monospecies and consortia were performed.

2017). Our model was trained on absolute abundance estimated from OD600 measurements and used to predict absolute abundance based on OD600 and thus automatically accounts for any potential biases.

To infer microbial interactions, time-resolved measurements of all monospecies and pairwise communities (66 combinations) were performed using an approximately 1:1 initial abundance ratio based on OD600 values (PW1 dataset, Appendix Fig S1, Dataset EV1). Monospecies growth and community composition were measured using OD600 measurements of total biomass and multiplexed 16S rRNA gene sequencing, respectively (see Materials and Methods). The monospecies displayed a broad range of growth rates, carrying capacities and lag phases (M dataset, Appendix Fig S2). Pairwise consortia exhibited diverse growth responses and dynamic behaviors including coexistence and single-species dominance (Appendix Fig S1). The distribution of absolute abundance of each species across communities in PW1 provided insight into variability in growth in the presence of a second organism (Appendix Fig S3A). Absolute species abundance was normalized to the monospecies maximum OD600 value to evaluate relative changes in the baseline fitness of each organism in the presence of second species. *Bacteroides* and CH displayed the lowest coefficient of variation (CV), indicating that the fitness levels of these organisms were not significantly modified by a second species (Appendix Fig S3B). The remaining species displayed a bimodal (FP, DP, PC, BH, and CA), long-tail distribution (ER and EL), and/or high CV (ER, BH, CA, and PC), demonstrating that growth was significantly altered in the presence of specific organisms.

To further probe the dynamic responses of pairwise consortia, a set of 15 consortia (Fig 2B, Dataset EV2) inoculated at different

initial species proportions based on OD600 values (95% species A, 5% species B, and the second wherein these percentages were reversed) were characterized using our experimental workflow (PW2 dataset, Appendix Fig S4). The community behaviors were classified into the following categories based on a quantitative threshold in species proportions at 72 h: (i) single-species *dominance*; (ii) *stable coexistence* wherein both species persisted above an abundance threshold; (iii) *history dependence* whereby communities inoculated using distinct initial species proportions mapped to different community structures; or (iv) *other* for communities that did not quantitatively satisfy the relative abundance thresholds for cases 1–3. A subset of the communities classified in the *other* category displayed weak history-dependent responses potentially attributed to variations in biological replicates. The qualitative behaviors of the remaining 51 pairwise communities were classified based on community structure at an initial ($t = 0$) and final ($t = 72$ h) time point using the PW2 experimental design wherein the organisms were inoculated using different initial species proportions (95% species A, 5% species B, and the reciprocal percentages, Appendix Fig S5A). Together, these results demonstrated that approximately 50, 24, and 12% of pairwise communities displayed dominance, stable coexistence, and history dependence, respectively (Appendix Fig S5B).

### Construction of a dynamic computational model of the community

A generalizable modeling framework was developed to infer parameters from time-series measurements of relative abundance and total

**Table 1.   Table of species used in study and associations with human diseases based on previous literature. Arrows pointing up or down denote positive or negative associations, respectively.**

| Species | Association(s) |
| --- | --- |
| *Prevotella copri* (PC) | Inflammatory and autoimmune disease (↑) (Scher *et al*, 2013), autism (↓) (Kang *et al*, 2013) |
| *Bacteroides vulgatus* (BV) | Ulcerative colitis (↑) (Bamba *et al*, 1995) |
| *Bacteroides uniformis* (BU) | Metabolic/immunological dysfunction (↓) (Gauffin Cano *et al*, 2012) |
| *Bacteroides ovatus* (BO) | Type I diabetes (↑) (Giongo *et al*, 2011) |
| *Bacteroides thetaiotaomicron* (BT) | Ulcerative colitis (↑)(Bloom *et al*, 2011) |
| *Faecalibacterium prausnitzii* (FP) | Crohn's disease (↓) (Watterlot *et al*, 2008), inflammatory bowel disease (↓) (Segain *et al*, 2000), Celiac disease (↓) (De Palma *et al*, 2010) |
| *Blautia hydrogenotrophica* (BH) | Healthy human colon (↑) (Nava *et al*, 2012) |
| *Eubacterium rectale* (ER) | Type II diabetes (↑) (Larsen *et al*, 2010) |
| *Collinsella aerofaciens* (CA) | Colon cancer (↓) (Moore & Moore, 1995), rheumatoid arthritis (↑) (Chen *et al*, 2016) |
| *Eggerthella lenta* (EL) | Cardiac drug transformations (↑) (Haiser *et al*, 2013), Crohn's disease (↑) (Thota *et al*, 2011), rheumatoid arthritis (↑) (Chen *et al*, 2016) |
| *Desulfovibrio piger* (DP) | Regressive autism (↑) (Finegold *et al*, 2012) |
| *Clostridium hiranonis* (CH) | None reported |

biomass (OD600). The generalized Lotka–Volterra (gLV) model represents microbial growth, intra-species interactions, and pairwise inter-species interactions and can be used to predict the dynamic behaviors of the community and analyze system properties such as stability and parameter sensitivity. The model equations are given by:

$$\frac{\mathrm{d}x_i}{\mathrm{d}t} = x_i\left(\mu_i + \sum_{j=1}^{n} \alpha_{ij}x_j\right),$$

where $n$, $\mu$, $\alpha_{ii}$, and $\alpha_{ij}$ represent the number of species, growth rates, intra-species, and inter-species interaction coefficients, respectively. To minimize overfitting of the data, a regularized parameter estimation method was implemented that penalized the magnitude of the parameter values (see Materials and Methods). Three training sets were evaluated based on predictive capability: (T1) M; (T2) M, PW1; and (T3) M, PW1, PW2. A range of regularization coefficient values ($\lambda$) was scanned to balance the goodness of fit to the training sets and degree of sparsity of the model (Appendix Fig S6). The parameterized gLV model trained on T3 captured the majority of pairwise community temporal responses (Fig 2A and B). However, the model did not accurately represent the dynamic behaviors of a set of communities including BH, EL; PC, CA; BO, CH; ER, BH; and PC, BH based on a threshold in the mean squared error between the model and data.

Thresholding the magnitude of the inter-species interaction coefficients using a value of 1e-5 yielded a densely connected network whereby 77% of species pairs exhibited an interaction. The network connectivity varied between 75 and 79% for interaction coefficient

thresholds ranging from 1e-6 to 1e-3. Interaction coefficients with a magnitude less than 1e-3 are not expected to change the steady state species abundance based on the inferred growth rate and interaction coefficient values. Of these interactions, 56 and 21% were negative and positive, respectively (Fig 2C). Negative interactions can arise from resource competition, biomolecular warfare, or production of toxic waste by-products. Positive interactions can originate from metabolite secretion or detoxification of the environment. *Bacteroides* (BO, BV, BU, and BT) displayed a net negative impact on the network, whereas EL, BH, and CH positively stimulated a large number of species (Appendix Fig S7A). Pairwise networks were enriched for unidirectional negative (−/0, 36%), bidirectional negative (−/−, 32%), and positive and negative (+/−, 26%) species couplings (Appendix Fig S7B). The contribution of each species to full community assembly in the model is dictated by a set of coupled ordinary differential equations that are a function of the monospecies growth rates, intra-species interactions, and outgoing and incoming inter-species interactions (see Materials and Methods). Therefore, a prediction about the role of each organism in community assembly requires simulation and analysis of the gLV model.

FP was the recipient of five positive interactions, suggesting that the fitness of FP is coupled to the composition of the community (Appendix Fig S7A). To determine the contribution of each incoming positive interaction on FP abundance, we examined a 6-member gLV model composed of FP, BH, BU, BV, CH, and DP. The combined set of five positive inter-species interactions was required to alter FP abundance by more than twofold, and single and dual inter-species interactions moderately increased FP abundance at 72 h (Appendix Fig S7C). Therefore, FP represents an ecologically responsive organism that is significantly enhanced by the presence of multiple organisms in the community in these conditions. Corroborating this notion, FP exhibited significant variability in absolute abundance across PW1 communities and frequent coexistence with other organisms (Appendix Figs S3 and S5B).

Strong positive or negative interactions can be deciphered by an enhancement or reduction in community productivity compared to a null model representing the sum of monospecies productivities (Foster & Bell, 2012). We computed the integral of biomass (OD600) over time to evaluate monospecies and pairwise community productivities (Appendix Fig S8). Our results showed that the productivities of 38% of pairwise consortia were less than twofold compared to the predictions based on the null models, consistent with the prevalence of negative interactions. *Bacteroides* pairwise communities BV, BU; BV, BO; BT, BU; BU, BO; BT, BV; and BT, BO exhibited significantly lower pairwise productivities compared to the null model, which was consistent with the inferred mutual inhibitory network topologies (Fig 2C). The productivities of EL, BH; EL, BU; EL, BT; EL, BO; EL, BV; and CH, ER were significantly enhanced in comparison with the predictions based on the null model. In the inferred network, five of these consortia (EL, BU; EL, BT; EL, BO; EL, BV; and CH, ER) displayed coupled negative and positive interaction topologies and EL, BH exhibited mutualism, demonstrating that both unidirectional and bidirectional positive interactions can augment community productivity.

Hierarchical clustering of the gLV interaction coefficients showed that members of *Bacteroides* or *Actinobacteria* exhibited similar

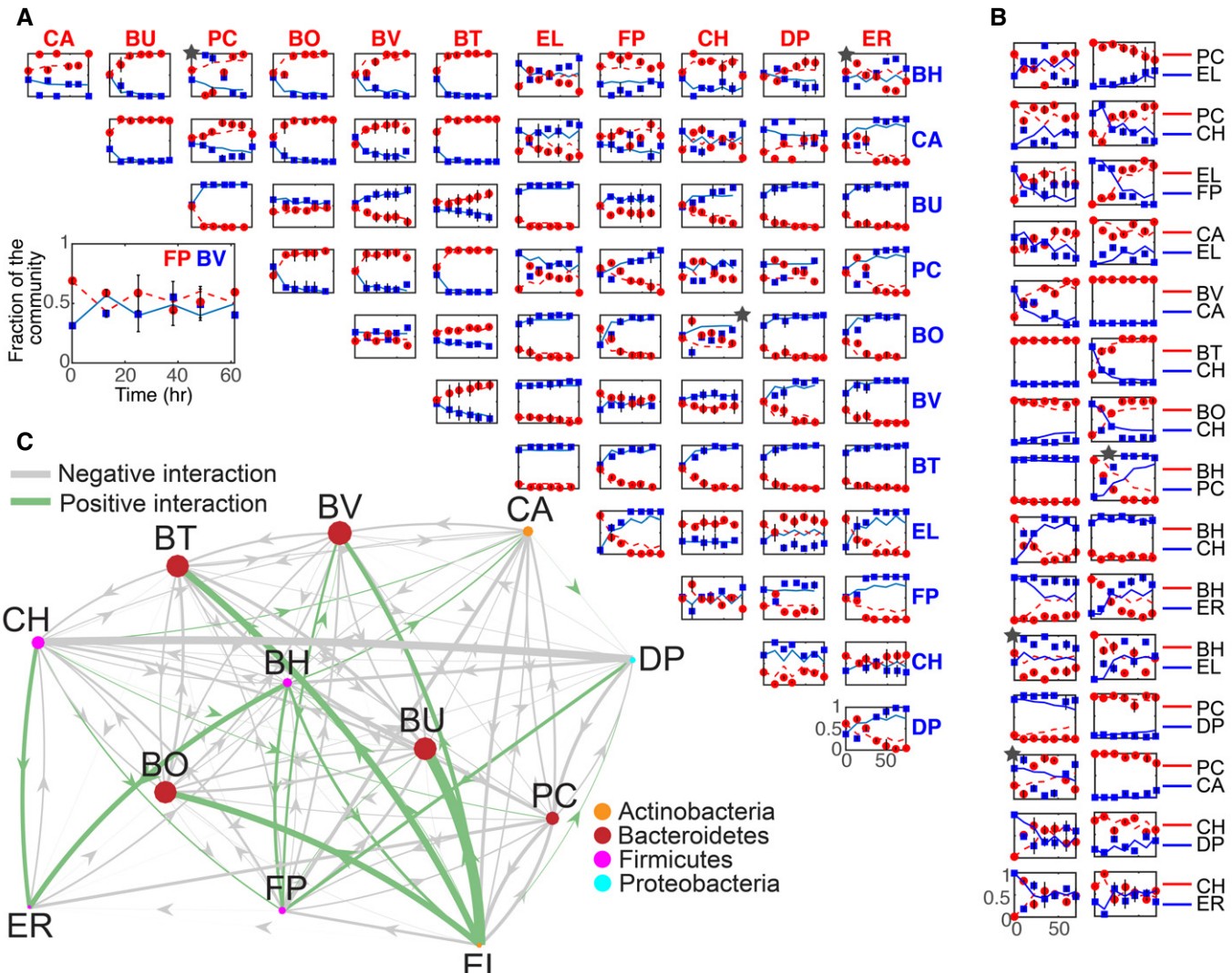

**Figure 2.  Model training of generalized Lotka–Volterra (gLV) to time-resolved measurements of monospecies and pairwise assemblages.**

A  Species relative abundance as a function of time for all pairwise communities. Experimental measurements and model fits based on T3 are represented as data points and lines, respectively. In each subplot, time and species relative abundance are displayed on the *x*- and *y*-axis, respectively. Stars denote datasets with a sum of mean squared errors greater than 0.15. Error bars represent 1 s.d. from the mean of at least three biological replicates.

B  Temporal changes in species relative abundance of a selected set of pairwise assemblages inoculated at 5% species A, 95% species B or 95% species A, 5% species B based on OD600 values. Time and relative abundance are represented on the *x*- and *y*-axis, respectively. Data points and lines represent experimental measurements and model fits to T3, respectively. Error bars represent 1 s.d. from the mean of at least three biological replicates. Stars denote datasets with a sum of mean squared errors greater than 0.15.

C  Inferred inter-species interaction coefficients for the gLV model trained on T3. Gray and green edges denote negative ($\alpha_{ij} < 0$) and positive ($\alpha_{ij} > 0$) interaction coefficients. The edge width and node size represent the magnitude of the inter-species interaction coefficient and steady state monospecies abundance ($x_e = -\mu_i \alpha_{ii}^{-1}$), respectively. To highlight significant interactions, inter-species interaction coefficients with a magnitude less than 1e-5 were not displayed.

patterns in outgoing and incoming microbial interactions (Appendix Fig S9A and B). However, the *Firmicutes* clustering pattern did not reflect the phylogenetic relationships. Together, these data show that distantly related species can display similar microbial interactions (e.g., BH and DP or PC and CA) and closely related species can exhibit distinct interaction patterns (e.g., BH and ER; Fig 1A). Therefore, evolutionary similarity was not a global predictor of the patterns in the inferred microbial interactions (Gómez *et al*, 2010; de Vos *et al*, 2017).

**Model validation using informative multi-species assemblages**

To validate the gLV model based on training set T3, time-resolved measurements of relative abundance of the full (12-member) and all single-species dropout communities (11-member consortia, Fig 3A, Dataset EV3) and total community biomass (Appendix Fig S10A) were performed. BU contamination was detected in the BU- consortium at 47.5 h (0.4% of the community) and BU persisted in the community until 72 h (2% of the community). Therefore, although

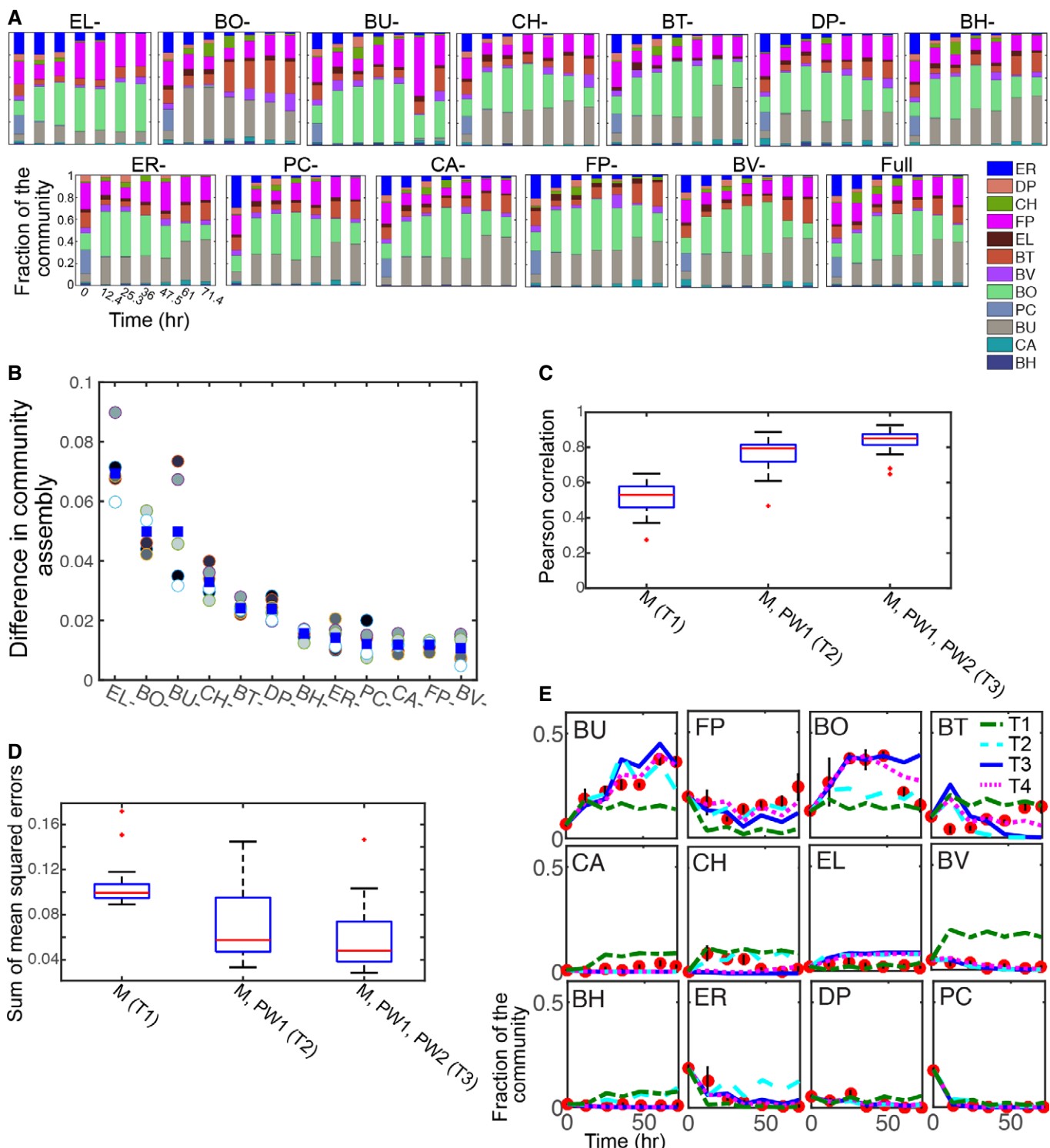

**Figure 3.**

the BU- consortium contained significantly lower amount of BU compared to the full community, BU was present for the final three time points. The community productivity computed by the integral of the growth response was significantly lower in the absence of EL (Appendix Fig S10B). To determine whether the synthetic

community could recapitulate the abundance pattern observed across human gut microbiome samples, we compared the abundance profiles in the synthetic community at 72 h and median relative abundance across human samples in a previous metagenomics sequencing study (Balzola *et al*, 2010). The median relative

**Figure 3.  Validation of the parameterized Lotka–Volterra model to time-series data of informative multi-species communities.**

A   Stacked bar plots of all 11 and 12-member (full) multi-species communities. The text above each subplot denotes the absent organism in the community. Time and relative abundance are represented on the *x*- and *y*-axis of each subplot, respectively. Colors represent different organisms in the community.

B   Difference in community assembly for single-species dropout communities. Difference in community assembly was computed as $\sum_{i=1}^{11} \frac{1}{7} \sum_{j=1}^{7} \left( \hat{v}_{ji,\text{FULL}} - v_{ji,X} \right)^2$ where $i$ and $j$ represent species and time points, respectively. $X$ represents a single-species dropout community lacking the organism $X$. $\hat{v}$ denotes the renormalized relative abundance of the shared set of species in the full (12-member) and single-species dropout community. Here, $\hat{v}_i = \frac{v_i}{\sum_{i \in C} v_i}$ where $C$ denotes the set of 11 shared organisms in the full and single-species dropout community. Data points represent biological replicates ($n = 6$, circles) and the mean of biological replicates (blue squares), respectively.

C   Box plot of the Pearson correlation coefficient for three model training sets (left) including T1: monospecies (M); T2: M and pairwise 1 (PW1); and T3: M, PW1 and pairwise 2 (PW2). On each box, the red line represents the median, the edges of the box are the 25th and 75th percentiles, the whiskers extend to the most extreme data points, and the outliers are plotted as red crosses.

D   Box plot of the sum of mean squared errors for the model predictions of the time-resolved measurements of multi-species communities trained on T1, T2, or T3 training sets. On each box, the red line represents the median, the edges of the box are the 25th and 75th percentiles, the whiskers extend to the most extreme data points, and the outliers are plotted as red crosses.

E   Comparison between model predictions that trained on T1–T4 and experimental data for each organism in the full community as a function of time. On each subplot, the *x*- and *y*-axis represent time and relative abundance of each species, respectively. Data points (red) and lines denote experimental data and model predictions based on T1–T4. Species names are displayed in the upper left corner of each subplot. Error bars represent 1 s.d. from the mean of six biological replicates.

abundance of seven species in the synthetic community that were present in the metagenomics dataset and mean relative abundance in the synthetic community were correlated ($\rho = 0.79$, $P < 0.05$), suggesting that microbial growth parameters and inter-species interactions are major variables influencing community structure in the human gut microbiota (Appendix Fig S11).

Characterization of the temporal variation in community structure in the presence and absence of each organism is a top-down approach to elucidate the contributions of single species to community assembly. The influence of each organism in community assembly can be evaluated by computing the difference in community dynamics between the single-species dropout and a renormalized full community that includes only the shared set of species. The absence of EL, BO, or BU significantly altered community assembly compared to the full community, whereas elimination of the remaining organisms demonstrated a moderate change in community dynamics (Fig 3B). The Pearson correlation between the time-resolved relative abundance of the single-species dropout consortium and the renormalized full community also highlighted EL, BO, and BU as influential organisms in community assembly (Appendix Fig S12A). The consortium lacking EL (EL-) exhibited a lower species diversity at later time points scored by Shannon equitability index $E_H$ (see Materials and Methods) compared to the predicted diversity computed based on a renormalized full community, indicating that EL promotes community diversity (Appendix Fig S12B). The consortium lacking BO displayed a higher diversity compared to the predicted values, indicating that BO reduces community diversity. The consortia lacking BU or CH exhibited a transient decrease in diversity that recovered to the null model set point by 72 h. The carrying capacity of the EL community was 28, 45, and 46% lower than the full community in the first, second, and third growth stage, illustrating that EL contributes significantly to community productivity (Appendix Fig S10B). Further, the growth rate of the EL consortium was reduced by 33 and 19% in the first and third growth stage compared to the full community. The impact of EL on community dynamics and productivity was disproportionately related to its stable low abundance in the full community, in contrast to the influential and highly abundant organisms BO and BU (Fig 3A and B). The species impact score, defined as the species relative abundance in the full community at 72 h plus the sum of the outgoing gLV inter-species

interaction coefficients, was correlated to the difference in community dynamics for single-species dropouts ($\rho = 0.67$, $P < 0.05$, Appendix Fig S13), and EL was the major driver of this correlation. These data suggest that the inferred ecological network and relative abundance pattern could explain the contribution of EL to multi-species community assembly.

The predictive capabilities of the parameterized gLV models were evaluated using the time-series measurements of multi-species communities. Validation metrics included the Pearson correlation coefficient between the model prediction and relative abundance measurements and the sum of mean squared errors of relative abundance across all species and time points in each community (see Materials and Methods). The model trained on T1 (M) exhibited the lowest median Pearson correlation coefficient $\rho$ (median $\rho = 0.53$) and largest error compared to the models trained on T2 (M, PW1) and T3 (M, PW1, PW2; Fig 3C and D), demonstrating that monospecies growth was not predictive of multi-species community assembly. The addition of PW1 to model training significantly increased the predictive capability of the model (median $\rho = 0.79$), highlighting that pairwise interactions are major variables driving community dynamics. The inclusion of PW2 (T3 training set) increased the predictive capability of the model (median $\rho = 0.85$) and reduced the error compared to the model trained on T2, indicating that the temporal responses of communities inoculated using distinct initial species proportions were informative for inferring model parameters.

The predicted monospecies OD600 based on the model trained on T3 and the fraction of each species in the full community at 72 h were not correlated ($\rho = 0.46$, $P = 0.13$), corroborating that monospecies growth failed to forecast the structure of the full community (Appendix Fig S14A). FP exhibited low monospecies fitness (Appendix Fig S2) and the second highest abundance level in the full community at 72 h, which was consistent with a large number of positive incoming interactions in the gLV model trained on T3 (Fig 2C, Appendix Fig S7A and C). By contrast, BV persisted at low abundance in the full community (Fig 3A) and exhibited high steady state monospecies abundance, consistent with a large number of inhibitory incoming inter-species interactions (Appendix Figs S7A and S14A). Therefore, the ecological network provided insight into variations in species fitness in the absence and presence of the community.

**Figure 4.   History dependence and robust species coexistence in pairwise consortia motifs.**

A   History dependence due to slow relaxation to equilibrium can be augmented by negative inter-species interactions. Model analyses of six pairwise communities that experimentally displayed history-dependent behaviors (Appendix Fig S5B) and are coupled by mutual inhibitory interactions in the gLV model trained on T3. Network topology (left) and heat-map of history-dependent responses (right) across a range of inter-species interaction coefficient values. The line width and node size of the network diagrams represent the magnitude of the inter-species interaction coefficients and steady state monospecies abundance, respectively. The heat-map shows the absolute value of the difference in species absolute abundance at 72 h for communities simulated using two initial conditions: $x_1 = 0.0158$, $x_2 = 0.0008$ or $x_1 = 0.0008$, $x_2 = 0.0158$ using the serial transfer experimental design shown in Fig 1B. The black box denotes the parameter regime for bistability in the model. The circle (red) indicates the inferred parameters based on training set T3.

B   Coupled positive and negative interactions can display robust species coexistence to variations in model parameters. Network diagram (inset) represents the magnitude, sign, and direction of the inferred inter-species interactions between CH ($x_1$) and ER ($x_2$). Dashed (gray) and solid (orange) lines indicate a positive and negative interaction, respectively. The line width and node size denote the magnitude of the inter-species interaction coefficients and steady state abundance of the monospecies, respectively. Heat-map of the ratio of $x_1$ (CH) to $x_2$ (ER) at 72 h as a function of the inter-species interaction coefficients $\alpha_{12}$ and $\alpha_{21}$. Initial conditions for simulations were $x_{1o} = 0.0008$, $x_{2o} = 0.0158$. The circle (black) indicates the inferred parameter values for the CH, ER consortium for the gLV model trained on T3.

C   Heat-map (right) of the ratio of $x_1$ to $x_2$ at 72 h across a broad range of growth rate parameter values ($\mu_1$ and $\mu_2$). Initial conditions for simulations were $x_{1o} = 0.0008$, $x_{2o} = 0.0158$. The line (white) outlines the parameter regimes for coexistence and single-species dominance at steady state. The circle (black) represents the inferred parameter values for the CH, ER consortium.

D   Box plot of the fraction of parameter space that exhibits species coexistence across a range of simulated growth rate parameters for all inferred positive/negative (+/−) or bidirectional negative (−/−) networks for the gLV model trained on T3. Two thousand five hundred combinations of growth rate parameters ranging from 0.05 to 1 h$^{-1}$ were evaluated. On each box, the red line represents the median, the edges of the box are the 25$^{th}$ and 75$^{th}$ percentiles, the whiskers extend to the most extreme data points that the algorithm considers not to be outliers, and the outliers are plotted as red crosses.

To evaluate the predictive capability of the model, the quantitative relationship between the model and data was examined for each species as a function of time using the parameter estimate deduced using T3 (Appendix Fig S14B). Across the majority of time points and multi-species consortia, the Pearson correlation coefficient between the model prediction and experimental data explained a high fraction of the variance and was statistically significant. While the model captured the temporal responses of the majority of species, the model did not reproduce the behavior of BT in the multi-species consortia and deviated at specific time points for BO and CH (Fig 3E, Appendix Fig S14B).

Incorporating additional datasets into the model training procedure could reduce the contributions of noise to parameter estimates and provide additional parameter constraints. To this end, the model was trained on M, PW1, PW2 and the time-resolved measurements of the full community (training set T4) and was validated on the 12 single-species dropout communities (Appendix Fig S15). The model predictive capability trained on T4 explained 81% of the variance ($\rho = 0.9$) on average of the multi-species community temporal responses. Model parameter sets based on T3 and T4 were highly correlated with the exception of a small number of interaction coefficients (Appendix Fig S15B). Over half (55%) of the interaction coefficients that were present or absent in the inferred network based on T4 compared to T3 involved the ecologically influential species EL and BO (Appendix Fig S15C). In sum, the pairwise gLV model could accurately predict the temporal changes in the majority of species in the multi-species communities, which suggests that higher-order interactions played a minor role in driving community assembly.

**History dependence and robust coexistence in pairwise assemblages**

We analyzed the model to elucidate the origins of history-dependent behaviors in pairwise communities in our experimental data (Appendix Fig S5). The model trained on T3 indicated that six of the eight pairwise communities that exhibited history-dependent behaviors in our experiments were linked via bidirectional negative

couplings and the remaining two networks displayed unidirectional negative interactions (Appendix Fig S5B). History dependence can arise from bistability or slow relaxation to a monostable equilibrium in the model. Bistability is a property of a dynamical system whereby the system has two stable steady states and can exhibit long-term dependence of the state of a system on its history, referred to as hysteresis. Bistability is a possible outcome of a gLV model of pairwise mutual inhibition (Murray, 2002).

To understand the origins of history dependence and contributions of model parameters to system behavior, we analyzed the inferred pairwise models of specific communities that exhibited history dependence in our experimental data (Fig 4A). In five of the six mutual inhibition networks, the consortia were not operating within a bistable parameter regime, indicating that the observed history-dependent responses stemmed from a slow relaxation to steady state (Fig 4A). The inferred parameter set for the bistable BU, BT consortium was located on the boundary between monostable and bistable parameter regimes, demonstrating that bistability was not robust to parameter variations (Fig 4A). History-dependent responses were evaluated by computing the difference in species concentrations at 72 h between models simulated from two different initial conditions: 95% species A, 5% species B, and the reciprocal condition. The models displayed history dependence across a range of inter-species interaction coefficient values beyond the bistable parameter regime for at least 72 h.

To investigate the physiological significance of such history-dependent behaviors, we computed the Euclidean distance of species concentrations from the equilibrium point across a broad range of simulated serial dilution rates (Appendix Fig S16). The rate of serial dilution represents colonic transit time, a major variable shown to modulate human gut microbiome composition, diversity, and functions (Roager *et al*, 2016; Vandeputte *et al*, 2016). Pairwise communities BO, BT; DP, PC; CA, PC; BO, BV; and BO, BU exhibited history-dependent responses up to 168 h due to inter-species interactions. In sum, our modeling results demonstrate that the time required to converge to steady state increases as a function of the dilution rate in monostable pairwise communities coupled by bidirectional or unidirectional negative interactions. Therefore, the time

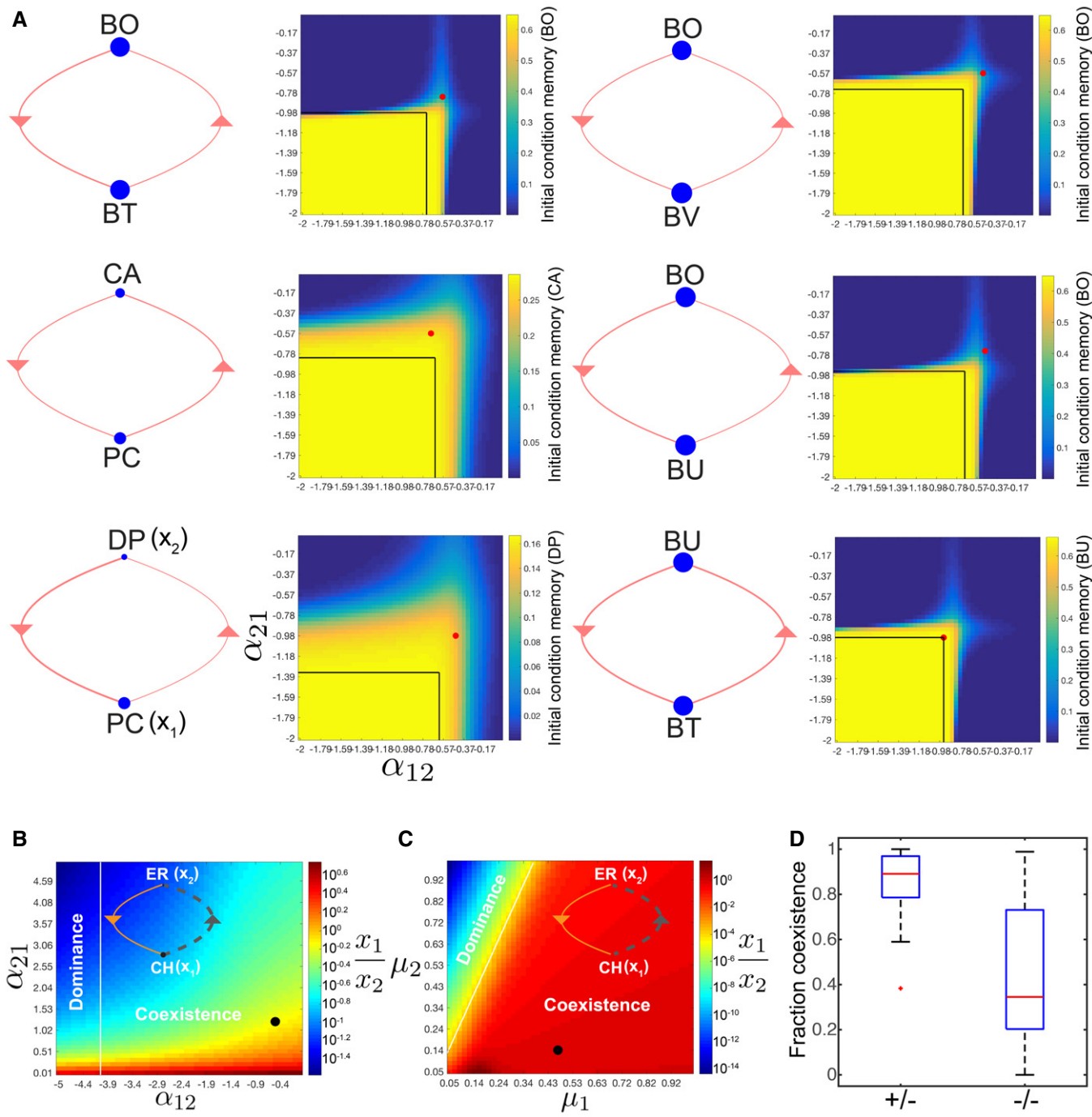

Figure 4.

required for communities to assemble to a steady state composition can vary over a broad range.

A subset of pairwise consortia exhibited stable coexistence wherein species inoculated at different initial proportions persisted in the community for duration of the experiment (Fig 2B, Appendix Fig S5). For example, the CH, ER pairwise community converged to an approximately equal abundance ratio as a function of time from distinct initial species proportions (Fig 2B). Stable coexistence is defined as nonzero steady state abundance of all

species and is a possible outcome of the generalized Lotka–Volterra model (Murray, 2002). We examined the network topologies and parameter dependence of stable coexistence in the model. The inferred pairwise network for the CH, ER community exhibited coupled positive and negative interactions (Fig 4B). The positive and negative interaction topology exhibits a broader parameter regime of coexistence compared to mutual inhibition (Fig 4B and C, Appendix Fig S17). The combination of positive and negative interactions establishes a negative feedback loop to modulate an

organism with higher monospecies fitness (e.g., CH), thus leading to stable coexistence with an organism that has lower monospecies fitness (e.g., ER) as a function of time.

Pairs of species that exhibited coexistence were linked by positive/negative and mutual inhibition in 50 and 25% of cases, illustrating that the positive/negative interaction motif was frequently associated with coexistence behavior (Appendix Fig S5B). The remaining species pairs that displayed coexistence encompassed unidirectional positive (12.5%), unidirectional negative (6.25%), and mutualism (6.25%). Across all inferred networks, the positive/negative interaction topology was significantly more robust to variations in growth rates compared to the bidirectional inhibition topologies (Fig 4D, Appendix Fig S18). We analyzed the network interactions at a higher taxonomic level to illuminate interaction patterns based on evolutionary relatedness. The inferred gLV interaction coefficients based on T3 were averaged across species associated with a given order since this level encompassed at least two species with the exception of DP in *Desulfovibrionales*. Several orders were connected by positive and negative interactions in the order-level interaction network, illustrating that the positive and negative interaction motif was consistent at higher taxonomic rankings (Appendix Fig S19).

### Analysis of model parameter constraint

Methods from Bayesian statistics can illuminate the uncertainty in the parameter values using the Posterior distribution, which represents the probability of the parameters given the data. Several microbiome modeling studies did not analyze the uncertainties in parameter estimates for gLV models, which provides information about whether the parameters are sufficiently constrained by the data or identifiable (Mounier *et al*, 2008; Stein *et al*, 2013; Buffie *et al*, 2015). To this end, the Metropolis–Hastings Markov chain Monte Carlo (MCMC) method was implemented to randomly sample from the Posterior distribution. The Markov chain was initialized from the inferred parameter set based on T3, and a burn-in period of 100,000 iterations was implemented to exclude the initial set of samples that do not represent the steady state distribution of the Markov chain (see Materials and Methods). The coefficient of variation (CV) of 90% of parameters was less than 0.5 for 570,000 iterations (CV values greater than 1 indicate high variability), indicating that the parameters were constrained by the data (Appendix Fig S20A). Parameters that were present or absent in the model trained on T4 compared to T3 exhibited a higher median CV equal to 0.37 compared to the unchanged set that had a median CV equal to 0.21 (Appendix Figs S15B and C, and S20B). These data suggest that model training on T4 provided additional constraints for specific parameters that displayed larger uncertainty in parameter estimates in the parameter set based on T3.

Parameter identifiability was evaluated by computing the Pearson correlation between all 12,090 parameter pairs for the single Markov chain simulated for 570,000 iterations initialized from the inferred parameter set based on T3 (Appendix Fig S20C). Parameters that do not influence observable variables are non-identifiable due to practical or structural reasons and can be correlated with other parameters (Gábor *et al*, 2017; Appendix Fig S20D and E). Correlations between parameters could lead to trade-offs in parameter space due to parameter couplings and poor convergence rates to the stationary distribution. Approximately 7.1% of parameter pairs

displayed a Pearson correlation coefficient greater than 0.6 or less than −0.6, indicating that the majority of parameters could be distinguished. The absolute value of the parameters was negatively correlated with the median absolute value of the Pearson correlation coefficient (Appendix Fig S20F, $\rho = -0.47$, $P = 9.3\text{e-}10$), demonstrating that parameters with smaller magnitudes were more frequently correlated with other parameters. In the parameter estimation procedure, regularization will lead to the reduction in the magnitude of parameters that do not link to the observable outputs.

### Interrogating microbial environmental impact using conditioned media

The net environmental impact of a single species at a defined time point can be represented by conditioned media, which has been depleted for specific resources and contains secreted metabolites. In conditioned media, positive interactions may be indicative of transformations of media components into substrates that can be utilized by the recipient species or detoxification of the environment. Negative interactions may derive from depletion of key nutrients or production of toxic compounds. In some cases, multiple mechanisms can combine to yield a net positive or negative effect on growth. Environmental pH is a major variable that can influence microbial growth responses. In co-culture, the environmental pH may not reflect the pH of the monocultures due to differences in metabolite secretion and degradation in a community. For example, cross-feeding of metabolic by-products such as acetate in the gut microbiota is a prevalent mechanism that could alter environmental pH (Duncan *et al*, 2002; Wrzosek *et al*, 2013; Rios-Covian *et al*, 2015).

We investigated whether changes in the recipient organism growth responses in the presence and absence of conditioned media from a source organism could be used to map microbial inter-relationships (Appendix Fig S21). The conditioned media impact score $R_{CM}$ was defined as defined as the ratio of the cumulative sum of the recipient organism growth response over 30 h in 75% conditioned media to unconditioned media. An $R_{CM} > 1$ or $R_{CM} < 1$ indicated a positive or negative influence of the source organism on the recipient organism. To evaluate the contribution of pH to the conditioned media growth responses, pH-adjusted conditioned media was prepared by modifying the pH to match the value of the unconditioned media.

Several factors could lead to disagreements between $R_{CM}$ and the gLV inter-species interaction coefficients, including, for example, a difference in metabolite utilization and secretion patterns of an organism in the presence of a second species or enhanced resource competition due to depletion of key resources (Filkins *et al*, 2015). Nevertheless, 75% of conditions were in qualitative agreement with the sign of the inferred gLV interaction coefficients based on T3 (Appendix Fig S21). Of the interactions that showed qualitative disagreement, 53% had a magnitude less than 1e-5. Together, these data demonstrated that in a majority of cases the effects of conditioned media on the growth of a recipient organism could predict the signs of incoming microbial interactions.

### Elucidating metabolic capabilities of monospecies via exo-metabolomics

Exo-metabolomics profiling of 97 major metabolites was performed on monospecies to determine metabolite utilization and secretion

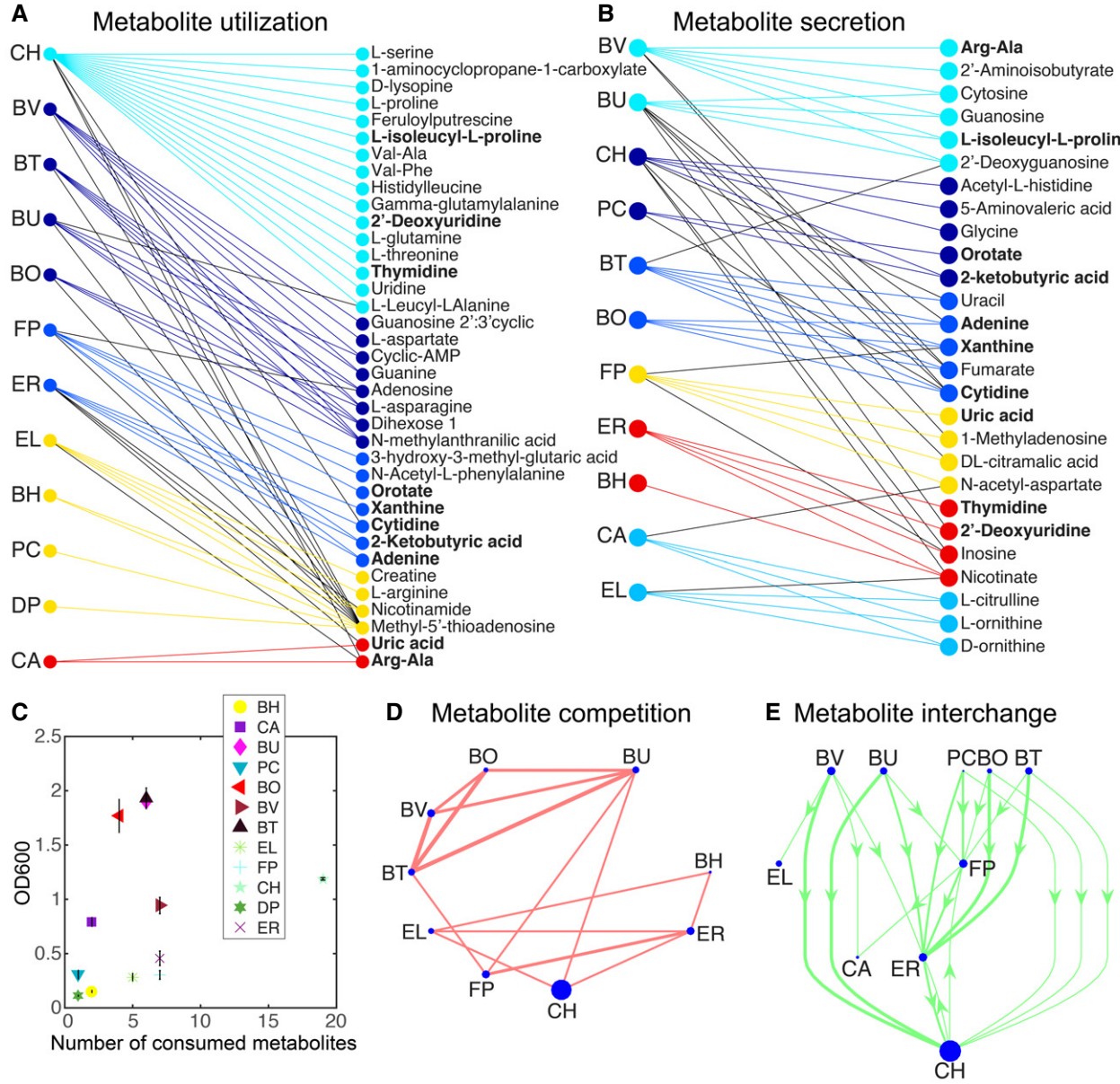

**Figure 5. Exo-metabolomics profiling of major metabolites elucidated the metabolic capabilities of monospecies.**

A    Bipartite network of species (left) and metabolites (right) for metabolites that decreased by at least twofold compared to the abundance of each metabolite at the beginning of the experiment. Colors represent modules containing many overlapping interactions in the network. Network partitioning into modules was performed using BiMat (Flores *et al*, 2015). Metabolites in bold were depleted and secreted by distinct organisms.

B    Bipartite network of species (left) and metabolites (right) for metabolites that increased in abundance by at least twofold compared to the beginning of the experiment. Metabolites highlighted in bold were depleted or secreted by different species. Colors represent modules containing many overlapping interactions in the network. Network partitioning into modules was performed using BiMat (Flores *et al*, 2015).

C    Scatter plot of the number of consumed metabolites that decreased by at least twofold compared to the beginning of the experiment vs. the OD600 value of the monospecies culture at the corresponding time point. Error bars represent 1 s.d. from the mean of three biological replicates.

D    Predicted resource utilization interaction network. Each edge represents at least two co-consumed metabolites that decreased by at least twofold, and the edge width is proportional to the number of co-consumed metabolites. Node size is proportional to the total number of consumed metabolites for each species.

E    Predicted metabolite interchange network representing metabolites that were secreted or utilized by distinct organisms based on a twofold threshold. Arrows point from the source species to the consumer organism. Node size and line width are proportional to the total number of secreted metabolites and number of predicted metabolite interactions, respectively. Species at the top and bottom of the network are primarily producers or consumers, respectively.

capabilities (Dataset EV4). The metabolite profiles were analyzed at an initial and final time point that occurred prior to 24 h to mirror the community experimental design with the exception of DP due to

insufficient accumulation of biomass for metabolomics measurements within the 24-h period (Fig 1B, Appendix Fig S22). A limitation of this study is that while significant cell lysis is not expected

for exponential or early stationary phase cultures, metabolites released from lysed cells would diminish apparent utilization and augment apparent secretion. Therefore, metabolite secretion is defined to operationally include potential cell lysis. Relative changes in metabolite abundances were computed using the log2 fold change of the final and initial time points, and a significance threshold of at least twofold was applied to the data (Fig 5A and B). We performed clustering analysis of the metabolite utilization and secretion networks to identify similarities in metabolite profiles. The clustering pattern did not recapitulate the phylogenetic relationships (Fig 1A) in many cases, demonstrating that distantly related species can occupy similar resource utilization niches (e.g., BH and EL or FP and ER) and closely related species can utilize distinct resources (e.g., BH and ER).

Our results showed a lack of correlation between the number of consumed metabolites and the total biomass produced by each monospecies at the corresponding time point. CH consumed the largest total number of metabolites in comparison with other organisms, thus representing a hub in the metabolite utilization network (Fig 5C). The total biomass produced by CH was not proportional to the total number of consumed metabolites or sum of log2 fold changes in utilized metabolites, suggesting that CH could be funneling energy toward cellular processes beyond biomass (Fig 5C and Appendix Fig S23A). Corroborating these results, CH exhibited a large number of positive and negative outgoing edges in the inferred inter-species interaction network (Appendix Fig S7A).

The sum of log2 fold change in metabolite secretion was correlated with total monospecies biomass ($R_s = 0.83$, $P < 0.002$), relative abundance in the full community at 72 h ($R_s = 0.66$, $P = 0.02$), number of outgoing negative interactions for each organism ($R_s = 0.65$, $P = 0.02$), and growth rate ($R_s = 0.74$, $P = 0.0008$), where $R_s$ represents the Spearman rank correlation coefficient (Appendix Fig S23B–E). These data suggest that metabolite secretion was a better predictor of species fitness and negative ecological interactions compared to metabolite utilization. The resource co-utilization network pinpointed significant resource competition between members of *Bacteroides* for a set of metabolites (Fig 5D). A network of predicted metabolite interchange illuminated producers (*Bacteroidetes*), consumers (EL and CA), and species that played dual roles (FP, ER, and CH). Metabolites predicted to mediate the largest number of pairwise negative interactions via resource competition included methyl-5′-thioadenosine (55 pairs), N-methylanthranilic acid (six pairs), and dihexose (three pairs) (Appendix Fig S24A). Metabolites implicated in three or more metabolite secretion interactions encompassed cytidine, adenine, and 2-ketobutyric acid (Appendix Fig S24B). FP was predicted to utilize several metabolites produced by BU, BO, PC, BT, and CH, illustrating a potential molecular basis of positive modulation by BU and CH in the inferred inter-species interaction network based on T3 (Fig 2C, Appendix Fig S7A and C). However, the metabolite profiles failed to predict the influential role of EL in mediating community assembly, fitness, and diversity (Fig 5D and E).

## Discussion

Developing the capabilities to predict microbial community dynamics is a first step toward elucidating the organizational principles of microbial communities and devising strategies for precisely manipulating ecological properties. The discovery of significant microbial inter-relationships and ecological driver species in the network can be exploited as novel control parameters for microbiomes. To this end, we developed a data-driven parameter estimation pipeline to build predictive dynamic models of microbial communities complementary to previously published methods (Bucci *et al*, 2016). In contrast to statistical network models, dynamic frameworks can be used to extract mathematical principles and probe system properties such as ecological stability, history dependence, and response to perturbations. Further, the inferred network can be used to define ecological roles for each species and model could be harnessed as a predictive tool for designing consortia with desired properties. We capitalized on methods from Bayesian statistics to go beyond a single parameter estimate to evaluate the uncertainty in parameters given the data and correlations between parameters. Future work will harness this information for experimental design by iteratively guiding the selection of informative experiments to reduce parameter uncertainties and thus enhance the predictive accuracy of the model.

Our results substantiate the notion that monospecies growth parameters and pairwise interactions dominate multi-species community dynamics (Friedman *et al*, 2017). It is possible that higher-order interactions significantly influence community dynamics in lower dimensional multi-species assemblages, such as three-member consortia. Previous work showed that pairwise phenomenological models of low-dimensional assemblages (2–3 species) trained on an interval of time of Monod-based community models failed in some cases to predict future dynamic behaviors (Momeni *et al*, 2017). The mechanistic models considered in this study involved a limited number of metabolites, whereas numerous metabolites likely mediate microbial interactions. Future work will explore the capability of pairwise models to recapitulate the dynamics of mechanistic models that capture such complexities. Here, we show that pairwise interactions can realize diverse behaviors encompassing history dependence, coexistence, and single-species dominance. Combinations of such interactions in multi-species assemblages can yield a diverse repertoire of dynamic behaviors and realize system-level properties including stability, resistance to invasion, and resilience to disturbances (Law & Morton, 1996; Mougi & Kondoh, 2012; Coyte *et al*, 2015).

We found that the time required for communities to assemble to a steady state composition can exhibit a broad distribution. Indeed, our results show that negative couplings in pairwise consortia can augment the time required to converge to a steady state community composition. Therefore, a steady state assumption for the human gut microbiome may not be valid if frequent environmental perturbations steer the system away from steady state on a faster timescale than the time required to relax to an equilibrium community composition (Bashan *et al*, 2016).

We interrogated an anaerobic synthetic ecology composed of prevalent human-associated intestinal species that play major roles in human health and disease (Table 1) and demonstrated frequent positive interactions and a large fraction of negative interactions. The network revealed hubs for negative (*Bacteroidetes*) and positive outgoing interactions (EL, CH, and BH). A top-down approach to investigate the contribution of species to community assembly pinpointed highly influential species EL, BO, and BU. These results

show that species with a large number of positive (EL) or negative (BO and BU) outgoing interactions can play an important role in community assembly. *Bacteroidetes* and *Firmicutes* have unique and complementary metabolic specializations in the gut microbiota (Fischbach & Sonnenburg, 2011), consistent with the numerous positive interactions deciphered between members of these phyla. *Bacteroides* and *Prevotella* have been shown to be anticorrelated across individuals (Arumugam *et al*, 2011; Ley, 2016). A recent study demonstrated that gnotobiotic mice colonized with BT and PC exhibited lower absolute abundance of both species compared to mono-colonized gnotobiotic mice, suggesting an inhibitory interaction (Kovatcheva-Datchary *et al*, 2015). In line with these results, BT excluded PC in pairwise experiments (Fig 2A, Appendix Fig S5A) and the inferred network showed that PC was inhibited by BT (Fig 2C). A previous study showed that BT negatively influences BV in metagenomics time-series data from one individual, consistent with inhibition of BV by BT in the inferred gLV network (Fig 2C; Fisher & Mehta, 2014).

Exo-metabolomics profiling identified CH as a hub for metabolite consumption and BU and CH as major producers of metabolites. Indeed, CH had the potential for significant environmental impact via utilization and secretion of a broad repertoire of metabolites and thus was predicted to have a significant impact on community assembly. However, CH did not persist at high abundance in the full community (Fig 3A and E) and had a moderate impact on community diversity and productivity (Appendix Figs S10 and S12B). The inferred network showed that CH was the recipient of a large number of negative interactions, which may limit the potential for influencing community functions (Appendix Fig S7A).

BU was a hub for metabolite secretion (Appendix Fig S23), which was consistent with its major role in shaping community assembly (Fig 3B). However, the patterns in metabolite utilization and secretion failed to explain the influential role of EL in community assembly. Our results highlight challenges in using environmental impact measured by exo-metabolomics to forecast microbial interactions and organism contributions to community functions. Co-consumption or secretion of a metabolite will generate a negative or positive interaction only if the substrate is limiting for growth. In addition, secondary metabolites or signaling molecules could contribute to the observed ecological relationships. Metabolite secretion as opposed to utilization was correlated to the number of negative inter-species interactions, suggesting that negative interactions may derive from mechanisms beyond resource competition such as biomolecular warfare or production of toxic metabolic by-products. Negative interactions due to resource competition or secretion of toxic compounds could lead to funneling of intracellular resources toward non-metabolic cellular processes such as stress, thus contributing to the lack of correlation between ecological interactions and resource utilization.

In the majority of cases, the effects of conditioned media on the growth response of a recipient species exhibited qualitative agreement with the sign of the inferred gLV interaction coefficient based on T3. As an organism grows, resources are depleted and toxic waste products accumulate, altering the energy availability of the environment. Therefore, spent media is a specific condition that may induce couplings between species that do not exist in communities growing in nutrient-dense environments. Corroborating this notion, approximately half of the conditions that showed qualitative

disagreement involved negligible interactions (Appendix Fig S21). As such, growth responses in conditioned media should be validated to enable accurate prediction of microbial interactions.

Microbial interactions have been probed in several synthetic ecologies of varying complexity and diversity including 18 *Streptomyces* strains (Wright & Vestigian, 2016), four-member freshwater isolates (Guo & Boedicker, 2016), and eight soil bacterial isolates composed of six strains from the *Pseudomonas* genus (Friedman *et al*, 2017). A significant fraction of the *Streptomyces* pairwise communities displayed frequent history-dependent and rare coexistence behaviors, which could be attributed to numerous mutual inhibitory pairwise interactions (Wright & Vestigian, 2016). By contrast, history-dependent responses were not detected in the synthetic consortium of eight soil isolates and coexistence was the prevalent community behavior (Friedman *et al*, 2017). On timescales of minutes, members of the freshwater isolate community did not display negative interactions that diminished cellular redox activity (Guo & Boedicker, 2016). Here, we interrogated the pairwise community dynamic behaviors in a 12-member anaerobic community spanning four distinct phyla and observed frequent positive interactions and stable species coexistence. Anaerobic metabolism has lower energy yields and requires the concerted activities of distinct community members to perform chemical transformations. Therefore, anaerobic microbial communities may exhibit a higher density of ecological inter-relationships and frequency of positive interactions compared to aerobic microbial communities. Such variations in the molecular mechanisms driving microbial interactions across distinct environments ranging from soil to the human gastrointestinal tract can manifest as differences in community-level properties including diversity, stability, and dynamic responses to perturbations.

Positive interactions were observed frequently in the synthetic human gut microbiome community in contrast to other synthetic communities (Foster & Bell, 2012). For example, EL had six positive outgoing interactions and displayed the largest impact on community assembly and productivity. These results show that positive interactions played a major role in shaping the dynamics and metabolic efficiency of the synthetic human gut community. Negative interactions have been shown to stabilize cooperative networks by introducing negative feedbacks (Coyte *et al*, 2015). Indeed, networks coupled by negative and positive interactions (+/−) contained 81% of the inferred positive interactions as opposed to mutualism (+/+, 9.5%) or unidirectional positive (+/0, 9.5%) topologies. The prevalent negative interactions in the inferred interaction network may promote ecological stability, whereas the frequent positive interactions modulate community assembly and metabolic efficiencies. Future work will elucidate the plasticity of the ecological network in response to changeable environments and the generalizability of these principles in higher-dimensional microbiomes that mirror the complexity of the natural system.

## Materials and Methods

### Starter culture inoculations

Cells were cultured in an anaerobic chamber (Coy Lab Products) using mixed gas tanks containing 85% $N_2$, 5% $H_2$, and 10% $CO_2$.

Starter cultures for community measurements were inoculated from 200 µl single-use 25% glycerol stocks into 15 ml of Anaerobic Basal Broth (ABB) media (Oxoid) in an anaerobic chamber and incubated at 37°C without shaking. To compare strains in similar growth phases that displayed variable lag phases, the strains were partitioned into slow and fast growth categories inoculated at 41 or 16 h prior to the beginning of the experiment, respectively. Strains in the slow growth category included *B. hydrogenotrophica* (BH, DSM 10507), *F. prausnitzii* (FP, DSM 17677), *C. aerofaciens* (CA, DSM 3979), *P. copri* (PC, DSM 18205), *E. rectale* (ER, ATCC 33656), and *D. piger* (DP, ATCC 29098). Fast-growing strains encompassed *B. uniformis* (BU, DSM 6597), *B. vulgatus* (BV, ATCC 8482), *B. thetaiotaomicron* (BT, ATCC 29148), *B. ovatus* (BO, ATCC 8483), *C. hiranonis* (CH, DSM 13275), and *E. lenta* (EL, DSM 2243).

## Microbial community culturing

Each constituent strain in microbial communities was inoculated at 0.01 OD600 unless otherwise noted. In multi-species assemblages, the total initial OD600 was 0.01 OD600 × $n$, where $n$ represents the number of strains in the community. For PW1, each component was normalized to OD600 of 0.01 and then mixed in equal proportion. For PW2, the major and minor strains were inoculated at 0.0158 and 0.0008 OD600, respectively. The microbial communities were arrayed using a liquid-handling robot (Biomek) into 96-well deep-well plates covered with a gas-permeable seal (Breathe Easy) and incubated at 37°C without shaking. In parallel, aliquots of the communities were transferred into a 384-well absorbance plate and grown in a Tecan Infinite F200 Pro plate reader with shaking for OD600 measurements at 30-min intervals. Samples from the deep-well plate were collected approximately every 12 h for a total of 72 h. At each time point, samples were mixed and 400 µl was transferred to a 96-well collection plate. The collection plate was centrifuged at 4,000 $g$ for 10 min, and 380 µl of the supernatant was removed with a multichannel pipette. Serial transfers were performed at 24-h intervals into fresh media using a 1:20 dilution. Species diversity was scored by the Shannon equitability index $E_H$ where $E_H = H \ln S^{-1}$ and $H = -\sum_{j=1}^{S} p_i \ln p_i$. Here, $S$, $H$, and $p_i$ denote the number of species in the community, Shannon diversity index, and relative abundance of the $i$th species, respectively.

## Genome extractions

Genomic DNA (gDNA) extractions were performed using the QIAamp 96 DNA QIAcube HT Kit (Qiagen) with minor modifications including an enzymatic lysis pre-treatment step and the use of a vacuum manifold to perform column purification steps. Enzymatic lysis was performed as follows: Cell pellets were resuspended in 180 µl of enzymatic lysis solution containing 20 mg/ml lysozyme (Sigma-Aldrich), 20 mM Tris–HCl pH 8 (Invitrogen), 2 mM EDTA, and 1.2% Triton X-100. Samples were incubated at 37°C for 30 min with shaking. Following this step, 4 µl of 100 ng/µl RNAse A (Qiagen) was added and samples were incubated at room temperature for ~1 min prior to administering 125 µl of proteinase K to buffer VXL (Qiagen). Samples were incubated for an additional 30 min at 56°C with shaking, 325 µl of ACB buffer was added, and the samples were transferred to a 96-well column plate for purification.

Samples were washed with 600 µl of AW1, AW2, and ethanol (Sigma-Aldrich). Following the ethanol wash, samples were allowed to dry for approximately 5 min. Finally, the gDNA samples were eluted using AE buffer pre-warmed to 56°C into a 96-tube rack plate and stored at −20°C.

## Illumina primer design, library preparation, and sequencing

Dual-indexed primers were designed for multiplexed next-generation amplicon sequencing on Illumina platforms. Each 90–99 base pair (bp) forward and reverse primers consisted of an indexed 5′ Illumina adaptor, heterogeneity spacer (Fadrosh *et al*, 2014), and 3′ annealing region to amplify 466 bp of the V3–V4 variable region of the 16S rRNA gene. The set of 64 unique forward (6 bp) and reverse (8 bp) indices allowed multiplexing of 1,536 samples per sequencing run. Oligonucleotides (Integrated DNA Technologies) were arrayed into 96-well plates using a stock concentration of 1 µM.

Following gDNA extraction, gDNA concentrations were quantified using the Quant-iT dsDNA High-Sensitivity kit (Thermo Fisher) and normalized to approximately 3 ng/µl. PCR amplification of the V3–V4 region of the 16S rRNA gene was performed with Phusion High-Fidelity DNA Polymerase (NEB) for 18–25 cycles using 0.05 µM of each primer. PCR amplicons were pooled by plate (96 conditions), purified (Zymo Research), and quantified using the Quant-iT dsDNA High-Sensitivity kit. The samples were normalized to the lowest sample concentration and then combined in equal proportions to generate the library. The library was quantified prior to loading using quantitative real-time PCR (KAPA Biosystems) on a CFX96 real-time PCR detection system (Bio-Rad). Following amplification, the library was diluted to 4.5 nM and loaded on the Illumina MiSeq platform for 300 bp paired-end sequencing.

## Data analysis pipeline for 16S rRNA gene sequencing

A reference database containing the V3–V4 16S rRNA gene sequences was constructed by assembling consensus sequences based on next-generation sequencing of monospecies cultures. To process the sequencing data, the MiSeq Reporter software demultiplexed the indices and generated the FASTQ files using the bcl2fastq algorithm. Custom Python 2.6.6. scripts were used for subsequent data processing steps and are available for download at Github (see Data Availability). First, paired-end reads are merged using PEAR (Paired-End reAd mergeR) v0.9.0 (Zhang *et al*, 2014). The global alignment tool in USEARCH v8.0 mapped each sequence to the reference database. A 97.5% alignment threshold was implemented to distinguish the closely related species *B. thetaiotaomicron* and *B. ovatus*. Relative abundance was computed by summing the read counts mapping to each organism divided by the total number of reads per condition. The data were exported for analysis in MATLAB (MathWorks).

## Conditioned media experiments

Strains were inoculated in 15 ml ABB according to the standard overnight culture inoculation protocol. To prepare the conditioned media, 10 ml of the source organism cultures was transferred to 50-ml Falcon tubes and filtered in the anaerobic chamber using

Steriflip (EMD Millipore). Ten milliliter ABB media was filtered as a control. Following pH measurements of the conditioned and unconditioned media, 5 ml was filtered a second time. The remaining 5 ml of each conditioned media was adjusted to the pH of the filtered ABB media using 1 M NaOH or 1 M HCl and sterile-filtered to represent the pH-adjusted condition. The response organisms were normalized to an OD600 of 0.04 in ABB media. A Biomek 3000 liquid-handling robot was used to transfer 60 μl of conditioned media into a 384-well plate (Corning), and 20 μl of the response organisms was added to a final OD600 of 0.01 in 75% conditioned media by volume. Plates were sealed (Diversified Biotech) and monitored every 30 min for 72 h in a Tecan Infinite F200 Pro plate reader at 37°C.

### Bacterial culturing for metabolomics

A single batch of ABB media was used for all steps in the metabolomics experiments. ER and FP were grown in ABB media supplemented with 33 mM acetate (Sigma). Overnight cultures were grown for 72 h at 37°C in 30 ml of ABB to saturation, diluted to 0.01 OD600, and aliquoted into three 30 ml replicates in a four-well plate (E&K Scientific). A well containing media was used to evaluate metabolite degradation as a function of time. Samples were collected immediately following cell inoculation.

At each time point, the wells were mixed via serological pipet prior to sample collection and 800 μl was removed for liquid chromatography–tandem mass spectrometry (LC-MS/MS). The samples were centrifuged at 6,000 *g* for 5 min, and 600 μl of supernatant was removed and filtered with a 0.22-μm filter unit (EMD Millipore). The supernatant was dispensed and simultaneously filtered into a microcentrifuge tube prior to freezing at −80°C. To elucidate metabolite profiles, samples were collected at different time points for each organism prior to 24 h except DP due to insufficient biomass accumulation for metabolomics measurements within the 24-h time interval.

### Exo-metabolomics measurements

Media samples (0.5 ml) were lyophilized until dry in a Labconco 6.5 L lyophilizer (Labconco, Kansas City, MO) and stored at −80°C until extraction. In preparation for LC-MS analysis, the dried samples were resuspended in 200 μl MeOH containing internal standards (25 μM 3,6−dihydroxy−4−methylpyridazine, 4−(3,3−dimethyl−ureido)benzoic acid, d5−Benzoic acid, 9−anthracene carboxylic acid, $^{13}$C−glucose, $^{13}$C−$^{15}$N−phenylalanine), bath sonicated for 20 min, then centrifuge-filtered through a 0.22 μm PVDF membrane (Pall) and placed into glass HPLC vials.

Liquid chromatography mass spectrometry (LC-MS and LC-MS/MS) was performed on extracts using a 1290 Ultra High Performance Liquid Chromatography stack (Agilent Technologies), with MS and MS/MS data collected using a 6550 Q-TOF Mass Spectrometer equipped with a dual AJS ESI source (Agilent Technologies). Hydrophilic Interaction Liquid Chromatography (HILIC) was performed using a SeQuant ZIC-pHILIC column with 5 μm particles at 200 Å porosity in 150 mm long and 2.1 mm internal diameter column housing (EMD Millipore). A ZIC-pHILIC guard column using the same stationary phase but in a 20 mm length housing was attached in-line at the column inlet.

The column was maintained at 40°C and solvent flow rate was kept at a constant of 0.25 ml min$^{-1}$ with a 2 μl injection volume for each sample. For each injection, the HILIC column was equilibrated with 100% mobile phase B (90:10 ACN:H$_2$O w/5 mM ammonium acetate) for 1.5 min, with a linear gradient to 50% mobile phase A (H$_2$O w/5 mM ammonium acetate) over 23.5 min, followed by a linear gradient to 65% A for 1 min with an isocratic hold at 65% A for 6 min; finally, the column was re-equilibrated with a linear gradient back to 100% B over 1 min, followed by an isocratic hold for 7 min. Samples were maintained at 4°C. All spectra were collected at a rate of two per second with the source gas at 290°C, drying gas at 11 l min$^{-1}$, nebulizer at 30 psig, and sheath gas at 200°C and 9 l min$^{-1}$; the fragmenter voltage was at 175 V. Full MS spectra were collected on individual samples using mass ranges of 30–1,200 m/z in positive mode and 60–1,200 m/z in negative mode. MS/MS fragmentation data was acquired using the autoMS1 function in separate runs on pooled samples with a mass range of 20–1,200 m/z in both polarities for both precursor and MS/MS scans with a 1.3 m/z isolation window, using 10, 20, and 40 V collision energies, 4 precursors per cycle with a 7,500 absolute abundance threshold and a target of 50,000 counts per spectrum. Exclusion of fragmented ions was set after collection of two spectra for 0.5 min. Metabolites were identified based on exact mass and retention time coupled with comparison of MS/MS fragmentation spectra to purchased standards.

LC-MS data was analyzed using the Agilent MassHunter Qualitative Analysis (Agilent Technologies) followed by the Metabolite Atlas workflow (Yao *et al*, 2015). A set of criteria was used to evaluate each of the detected peaks and assign a level of confidence in the compound identification. Compounds given a positive identification had matching retention time and m/z to a pure standard run using the same methods described above. A compound with the highest level of positive identification additionally had a matching MS/MS fragmentation spectrum to either an outside database (METLIN) or collected in house. Putative identifications were assigned to compounds with matching m/z and MS/MS spectrum.

### Model

The generalized Lotka–Volterra (gLV) model was used to represent microbial community dynamics. The gLV model is a set of coupled ordinary differential equations that represent the temporal variation in species abundance ($x_i$). The model equations are given by:

$$\frac{dx_i}{dt} = x_i\left(\mu_i + \sum_{j=1}^{n}\alpha_{ij}x_j\right),$$

where $n$, $\mu$, $\alpha_{ii}$, and $\alpha_{ij}$ represent the number of species, growth rates, intra-species interaction coefficients, and inter-species interaction coefficients, respectively. This model requires that the intra-species interaction coefficients are negative ($\alpha_{ii} < 0$). Inter-species interaction coefficients $\alpha_{ij}$ can be positive or negative, representing a stimulatory or antagonistic microbial interaction.

### Parameter estimation and validation

Custom scripts in MATLAB (MathWorks) were used for model analysis and parameter estimation. A generalizable parameter

estimation framework was developed to infer 156 ($n^2 + n$) parameters of the gLV model from time-series measurements of absolute species abundance. The experimental data included time-series measurements of OD600 for monospecies and communities and relative abundance of each species in the communities based on 16S rRNA gene sequencing. The initial conditions for model simulations were computed by multiplying the total initial biomass of each community by the species proportions at the first time point. It is challenging to infer model parameters from compositional data generated by next-generation sequencing since this is an underdetermined problem and there are many solutions that would yield the same relative abundance output (Fisher & Mehta, 2014). The absolute abundance of each species was estimated by the product of the relative abundance and the OD600 value (total biomass) at each time point. The nonlinear programming solver requires an initial point to the optimization problem. The initial point was computed by transforming the gLV system of equations into a linear system of equations by dividing the left and right side of the equation by $x_i$ and substituting $\frac{d \log(x_i)}{dt}$ for $\frac{1}{x}\frac{dx}{dt}$ (Mounier *et al*, 2008).

$$\frac{d \log(x_i)}{dt} = \mu_i + \sum_{j=1}^{n} \alpha_{ij} x_j$$

A linear least-squares algorithm (MATLAB) with bounds was used to solve for the unknown parameters using time-series monospecies and pairwise community measurements.

To minimize overfitting of the data, L1 regularization was used to penalize nonzero parameter values. Experiments were weighted equally in the objective function. A nonlinear programming solver (MATLAB) was used to minimize the following objective function:

$$F_y = \sum_{i=1}^{l} \sum_{j=1}^{m} \frac{1}{p} \sum_{k=1}^{p} (\hat{y}_k - y_k)^2 + \lambda |\Theta|$$

Here, $l$, $m$, and $p$ denote community datasets, species in each community, and time points, respectively. $\lambda$ and $\Theta$ represent the regularization coefficient and the parameter vector. $\hat{y}_k$ and $y_k$ designate the model prediction and experimental measurement of absolute abundance of a species at a specific time point $k$.

In the optimization problem, an optimal $\lambda$ value was identified to minimize overfitting of the data by balancing the goodness of fit and sparsity of the model. To do so, values of $\lambda$ were scanned from $10^{-5}$ to 10. An optimization method was implemented to infer the best estimate of the model parameters using $\lambda = 0.0077$. The goodness of fit of the model was computed using the following equation:

$$G_f = \sum_{j=1}^{m} \frac{1}{p} \sum_{k=1}^{p} (\hat{r}_k - r_k)^2,$$

where $m$, $p$, $\hat{r}$, and $r$ represent species, time points, predicted relative abundance, and measured relative abundance, respectively. Validation of the model was performed using $G_f$ and the Pearson correlation coefficient between the model prediction and experimental data of all species in each community at each time point. Inferred parameter sets based on training sets T1-4 are shown in Appendix Figs S25–S28.

## Parameter uncertainty analysis

Bayes rule introduces the notion of prior and posterior information:

$$P(\theta|y) = \frac{P(y|\theta)P(\theta)}{P(y)}$$

The parameters and data are represented by $\theta$ and $y$. We assume that the error is normally distributed with a mean of 0 using the following equation $y = f(\theta) + \epsilon$, where $\epsilon \sim N(0, \sigma^2)$. The Posterior distribution $P(\theta|y)$ represents the uncertainty in the parameters. Since direct sampling from the Posterior distribution is challenging, the Metropolis–Hastings algorithm was used to estimate this distribution. The Metropolis–Hastings algorithm was simulated for 670,000 iterations from the inferred parameter set based on training set T3. A burn-in period of the first 100,000 was excluded from the analyses to allow the chain to converge to the stationary distribution.

To evaluate convergence, four independent chains were simulated for 670,000 iterations using a burn-in period of 100,000 iterations from randomly sampled parameter values using a normal distribution with the mean equal to the parameter estimate based on training set T3 and standard deviation of 0.3 times the value of each parameter. The Gelman–Rubin potential scale reduction factor (PSRF) was used to evaluate convergence of the Posterior distribution estimate. If the chains have converged to the target Posterior distribution, the PSRF should be close to 1. Our results showed that 60% of parameters have a PSRF less than 1.5. The PSRF is negatively correlated to the parameter magnitudes based on training set T3 ($R_s = -0.73$, $P = 0$ where $R_s$ and $P$ represent the Spearman correlation and $P$-value, respectively), suggesting that poor convergence is isolated to a specific set of parameters that exhibit small magnitudes due to regularization and do not significantly impact model dynamics. For all five Markov chains, the mean of each parameter across 570,000 iterations was highly correlated to the parameter set based on T3 ($\rho \geq 0.99$), suggesting that MCMC was exploring a single mode of the Posterior distribution.

## Data and software availability

The code for analyzing community composition based on next-generation sequencing data is available at Github: https://github.com/ryanusahk/NextGenSequencingScripts_RH (doi: 10.5281/zenodo.1248186). Metabolomics data are available through the JGI Genome Portal (Project Id 1198552). Next-generation sequencing data are available through the European Nucleotide Archive (accession number PRJEB26607). Computational models (Code EV1) contain the Systems Biology Markup Language (SBML) code for simulating the generalized Lotka–Volterra models trained on T1–T4. Curated model files have also been made available at JWS online at https://jjj.bio.vu.nl/database/venturelli.

**Expanded View** for this article is available online.

## Acknowledgements

We would like to thank Nicholas Justice, Michael Fischbach, and Justin Sonnenburg for helpful discussions. We would like to thank Suzanne Kosina for help with metabolomics methods. We are grateful to Ryan Clark, James Papadopolous, and Joshua Hamilton for critical reading of the manuscript. S.H. was supported by the National Institute of General Medical Sciences of the

National Institutes of Health under Award Number T32GM008349. O.S.V. was supported by the Simons Foundation at the Life Science Research Foundation postdoctoral fellowship. This work was supported by the Defense Advanced Research Projects Agency (DARPA) Grant HR0011516183.

### Author contributions

OSV and APA designed the research. ACC, GF, RHH, and SH carried out the experiments. OSV designed and implemented computational modeling methods. OSV, ACC, RHH, and GF analyzed the data. OSV wrote the manuscript. ACC, GF, RHH, and APA assisted in revising the manuscript. TN, OSV, APA, and BPB designed metabolomics experiments, RL performed LC-MS measurements. RL and BPB assisted in analyzing the data.

### Conflict of interest

The authors declare that they have no conflict of interest.

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
