## [Review Process File · Molecular Systems Biology]

Deciphering microbial interactions in synthetic human gut microbiome communities

Ophelia S. Venturelli, Alex C. Carr, Garth Fisher, Ryan H. Hsu, Rebecca Lau, Benjamin P. Bowen, Susan Hromada, Trent Northen & Adam P. Arkin

Review timeline:

Submission date:	9 December 2017
Editorial Decision:	16 January 2018
Revision received:	16 April 2018
Editorial Decision:	20 April 2018
Revision received:	13 May 2018
Accepted:	22 May 2018

Editor: Maria Polychronidou

Transaction Report:

1st Editorial Decision

16 January 2018

Thank you again for submitting your work to Molecular Systems Biology. We have now heard back from the three referees who agreed to evaluate your study. As you will see below, the reviewers are overall quite positive. They raise however a series of concerns, which we would ask you to address in a revision of the manuscript.

The reviewers' recommendations are rather clear and therefore I think that there is no need to repeat all the points listed below. Please let me know in case you would like to discuss further any of the issues raised by the reviewers.

REVIEWER REPORTS

Reviewer #1:

This is an important study that studies interactions between gut microbes using assembled in-vitro communities. This approach obviously has its limits in that it is not in-vivo but it allows detailed study and modelling of defined communities and the inferences of interactions and other ecological properties. This is sorely lacking from the great majority of microbiome work, which often rests upon single time point analysis of in vivo communities. Just for methods alone then, this paper is a major step forward and in addition the authors present a number of interesting and important results that suggest that their approach is the right one. Key amongst these is the finding that pairwise interactions allow one to predict much of the interaction networks of larger communities. There are a lot of negative options out there on these approaches that rest upon the challenges that come with the potential higher order interactions in communities. These opinions often miss the fact that LV like models can certainly deal with this conditionality although it does make everything harder to predict. Moreover, the authors here show evidence that the biology is also on our side in finding that these higher order effects are not dominant in these systems. While this is only in vitro at this point,

this is a very promising and important step for the future of predicting these vital and complex communities. I have only a couple of other comments:

1. How reliable is OD as a measure of abundance? Cell adhesion, size and shape can all affect OD. Did the authors plate out cells for CFU and check it matches reasonably well? These are culturable bugs so it is certainly possible.
2. What is the basis for the decision of the media used to grow the communities? Given the importance of media for growth and interactions, I think some discussion of this would be helpful.
3. The authors choose to end by contrasting their results to our Science paper that argued that cooperation can be destabilising for microbial communities. While I realise that this has a nice rhetorical ring for a closing, I find this contrast a bit of a straw man as we never said that cooperation is not important for communities. Indeed, we go to some length discuss the importance of cooperation for things like productivity, which is a major ecological factor, and it is clear that if one species props up another, positive (one or two way) interactions can be important for diversity in a stable system. What we argued, however, is that too much cooperation can be a bad thing owing to the effects on stability, which many non-network-theory people found surprising. Looking at the authors' network in Figure 2 I am struck by the fact that a) the majority of interactions are negative and b) the majority of positive interactions appear to be exploitative (+/-) or commensal (+/0) not cooperative (+/+) as implied by the closing paragraph. (It would be good to have numbers on these interaction frequencies if they are not somewhere that I missed). This kind of network is exactly the kind that we predicted would display stability as there will be negative feedbacks in there. One does not need the system to have all negative interactions, one just needs a certain number of these feedback. Indeed, we show (admittedly in our supplement Figure S2) that adding exploitative interactions to a cooperative network can stabilise it, just like adding competitive interactions.

Kevin Foster

Reviewer #2:

The manuscript entitled "Deciphering microbial interactions in synthetic human gut microbiome communities" by Venturelli & Carr et al. describes a set of experimental and computational explorations of community assembly using a set of 12 bacterial species representing the diversity in the human gut microbiome. While there are many findings in the paper, the main finding put forward in the abstract is the importance of pairwise interaction in community formation.

This work is intriguing. The results of pairwise competitions and the synthetic community experiments, including dropouts, provide a nice resource for the microbiome community. The ability of the model to predict community formation so well from pairwise growth rates is exciting. While some works have used pairwise interactions to predict the composition of more complex communities (e.g. Friedman 2017), that simple models including only pairwise interactions perform well for human-associated microbes (e.g Fig 3d) is exciting.

This paper makes many claims throughout. A number of claims are not well supported by the data, require further clarification or refinement of methods (e.g. analysis of robustness to outliers), or are contradicted by other parts of the paper. In particular, the stability of the communities on a whole to dropouts (Fig 3a) appears to contradict the importance of pairwise interactions highlighted throughout, and this requires further explanation. The connections between different sections are not always clear, and the methods are not always well explained.

Major comments

The single-species dropouts revealed different organisms to be important to community assembly than suggested by the network hub analysis -- many species with high outgoing interactions have minimal effect in the dropout experiments. Can the authors elaborate further on this? At face value, this does not appear to be consistent with the claims that pairwise interactions are essential to explain most of the data.

The metric used to identify the effect of dropouts is most sensitive to changes in the abundance of the most abundant species. Perhaps an alternative metric could better capture, other, more subtle changes. A simple thing that could be done is to create a 'predicted' community for each dropout by subtracting the removed strain from the final full community composition, and renormalizing. Evenness could be compared to the predicted evenness, Pearson correlations could be calculated, or something else.

Interaction terms in these gLV models can be negative because of simple competition for resources (this is why self-self interaction is always negative). This should be stated plainly somewhere for the uninitiated reader, both in the results and in the discussion (e.g. around Line 463).

FP is a very slow grower in these conditions (Fig S2). Perhaps its discovery as a recipient species is dependent upon the particular conditions of the media? This may be wind up being biological (similar nutrients may be limiting in vivo), but the results may be entirely dependent on the media. Therefore, the claim that FP is ecologically responsive Line 176-178 is overstated (adding "under these conditions" in line could work).

Similarly, the mentioned absence of "context-dependent interactions" in the abstract is not supported, as pairwise interactions may be conditional to this media. Further, a lot of the community in Figure 3 is still not explained by the pairwise data, even in this media. Line 262, "conditional interactions played a minor role in driving community assembly", is also an overstatement. (Probably other locations as well)

It would be helpful to see lines in Figure 3d showing the predictions from the model without T2 and T3. This would make it easier to evaluate if the inclusion of the pairwise interactions gets much closer to reality than growth rates alone, and why.

Line 193-7: How much more predictive power does including the species interactions give than the growth parameters for comparison to metagenomic data?

Line 212-216: Does this analysis hold if EL is removed? It looks to be driving most of the signal in Figure S12.

Line 314-316: How more likely are +/- pairs than -/- pairs given the distribution of positive and negative interactions in the observed data set?

Line 314-316: How robust are the described disparities between the topologies to different sets of growth rates?

Modeling constraint/ MCMC modeling: I found this section weak, though I am not an expert in MCMC models.

A) In the methods, the authors state that "If the chains have converged to the target posterior distribution the PRSF should be close to 1" and then "Our results showed that 81% of parameters had a PRSF less than 3." Three is not that close to one, so I can't understand why this is used as evidence of convergence. The distribution across parameters (histogram) might also be useful.

B) Line 329-330: How meaningful is this given that the priors were set with a CV of .3?

C) Was the MCMC run in such a way that the sign of an interaction could change? How would this affect the results?

D) Sup Fig 18f: The connection between coefficient magnitude and constraint would be easier to evaluate if a scatter plot of average correlation vs average absolute effect size were shown.

E) Overall, the methods section is very sparse for this section, and it isn't always clear what is being talked about in the text. This is important, as the average reader of MSB does not have expert-level knowledge in these methods. For example, it is not clear in the text that the CV estimates are on the last 200,000 iterations of the chain or if the Pearson correlations are across or within MCMC chains. The legend of 18f refers to 'at least 25' Pearson correlations, but the Methods state that 4 MCMC chains were run. It is also not clear if the last sentence on page 11 is referring to some step in this MCMC constraint estimation or something else.

How much are metabolite secretion measurements affected by randomly lysed cells (e.g. during stationary phase), or cell lysis during sample preparation? Can you rule out that such artifacts are impacting the results? How does secretion correlate with growth rate? While secretion could be operationally defined to include cell lysis, this would change the interpretations made in the discussion (lines 496-502).

The metabolite analysis is interesting but not connected much to the observed species interactions. Are the interactions predicted by metabolites reflected in the model and/or the spent media experiments? Why or why not?

How does the analysis in Supplementary Figure 21a change if the outlier is removed?

Specific comments:

Growth rate of some slow growers (e.g. FP; CA) does not appear to be fast enough to maintain saturation in 1:20 dilutions of single cultures. How was carrying capacity then defined? The authors might refer to the number used for normalization as "max OD".

Line 184: This statement is quite strong and should be supported with some sort of quantitative argument or toned down. It is likely that the phylogenetic signal is quite important but changes quite rapidly, such that only bacteria related below a certain taxonomic level would have similar interaction patterns - and sufficient resolution was provided only within the genus level.

Error bars are needed in Figure S11.

Line 193-7: Why were only 7 species used for comparison to metagenomic data? Not enough information is given here.

Line 274: Does "history-dependent dynamic behavior in the model" mean that the data used for determining history-dependence were from the model predictions, not from experiments?

Line 318-321: The link between the network topology at this level and the observed stability of the gut microbiome is highly speculative. What justification is there to average interaction coefficients across members of a phylum? Why at this taxonomic level and not others? What about the other topologies in this network?

Line 23-5: This is highly speculative language. Colonization resistance is an emergent property, not quite a core function. Chemical transformations by microbes have some effects on the human host, but it is not clear that any currently known reaction is a "core function" of the microbiome.

Line 41-2: The cited paper did not show stability on the scale of decades.

Line 48-9: F/B ratio has not stood the test of time (<http://mbio.asm.org/content/7/4/e01018-16.full>)

Line 50-1: This statement is strong and very speculative given the state of the field.

Which strains were used for each organism?

Typos and polish:

A version of Figure S8 and S5 clustered by phylogeny would be helpful for interpreting claims of lack of phylogenetic clustering of interactions.

Fig S2: Data points and "shading" are not distinguishable in PDF

Line 170: Amensalism is jargon. While this word is defined in the following parentheses, it is not obvious in to a non-specialist reader what this notation means.

Line 232: Clarify what "the model" means here.

Line 552: What is a sequencing plate here?

Line 553: Breath -> Breathe

Fig 3c is not necessary for a main figure.

Figure 4c would be strengthened with a schematic graphic similar to that shown in 4b.

Line 314-316: I get this sentence, but it was extremely hard to parse.

Line 324-5: Please qualify this sentence to explain that you are talking about microbiome modeling papers. Do you mean that no microbiome modeling manuscripts have evaluated uncertainty, or just some?

Bipartate is jargon.

Descriptive text on the panels in Fig 5 would help orient readers

Legend for Fig 3d should state that model T3 is used here.

Parameter symbols aren't neatly introduced until the Methods, but are used in the main figures.

Perhaps remove symbolic representation from main figures or introduce in main text.

Fig S3: A line at $x=1$ would be helpful for interpreting this Figure.

Reviewer #3:

Despite the known health importance of the human gut microbiome, surprisingly little is known about how individual species within the gut interact and the consequences of these interactions for the structure of the community. As a first step to address these issues, the authors used a collection of 12 gut bacteria to make pairwise competition experiments, while tracking the relative abundances across time with multiplexed 16s sequencing, and the total biomass with light absorbance. Three data sets were constructed: the first one includes only the single species growing; the second one includes also the pairwise competition experiments starting at 1:1 ratio; and the third one adds up the competition experiments starting at 95-5% and 5-95%. These temporal datasets were used to estimate the parameters of a Lotka-Volterra model; the authors find that the pairwise interactions do a pretty good of predicting the outcome of a variety of multispecies competitions (better than the monoculture growth alone). This conclusion reinforces what previous recent studies have said about communities in other environments, but it is worth stressing the importance of studying gut microbes despite the extra difficulty of working anaerobically. Interestingly, the metabolomics provided surprisingly little insight into the interactions, which I view as a significant warning or challenge to the field. I think that this paper is a nice contribution and could be published with only minor changes.

The authors measure a number of positive interactions between species, and this is contrasted with the results of Foster & Bell (2012). However, it is important to note that Foster & Bell only measured total biomass (not individual fractions), meaning that a positive interaction in that paper actually requires a very strong (and likely two-way) positive interaction. If the authors analyzed their data in a similar way would they see any examples of an increase in total biomass? This is relevant because in our past work we saw many examples of one-way facilitation but no examples of two-way mutualisms in which the total biomass was increased (Friedman et al, 2017). More generally, I don't understand how the authors could observe so many positive interactions if a majority of the competitions lead to competitive exclusion.

We find that the species that gain from other species are typically those that grow very poorly; was this the case for *F. prausnitzii*, which had five incoming positive interactions?

As far as I understand, the sequencing of 16S was done every 12 hours, whereas the dilutions were done every 24 hours. Do the authors see examples of apparent "oscillations" in relative frequencies? We normally only make frequency measurements with a period that is equal to (or a multiple of) the dilution period because otherwise there can be oscillations that are in some sense an artifact (the enlarged time trace in Fig 2a may have such dynamics...).

Also, in Fig 1b the time trace coloring and the fraction bar plot coloring seem to imply something that is not true. The fractions in the bar plot are within a given well (that has a given color - either blue or green), whereas the bar plot has both colors shown.

I found the discussion of history-dependent (HD) pairs to be a bit confusing. Ideally, the authors could distinguish between slow relaxation to an equilibrium and bistability. I understand that this is not always possible, but I couldn't quite figure out what the authors think is happening with their pairs. We see some pairs that are probably slow relaxation, other pairs that show clear bistability

(with only one species surviving at equilibrium), other pairs that show bistability with one of the states corresponding to coexistence (which is not an allowed outcome of LV), and a few ugly pairs that are difficult to categorize. As a reader, I would have appreciated a better sense of how the authors history-dependent pairs break down among these categories.

I have particular concerns about the categorization that the authors are using, which involves parameterization of the LV model. Instead, I would much prefer it if the authors directly look at the data and tell us what outcome they think is happening (a LV parameterization of an outcome not allowed by LV model will likely be misleading).

I would have appreciated a separate figure on this topic where the authors show the raw data and with a description of what the authors think is happening.

I like the idea that the pairs with +/- interactions establish a negative feedback loop that leads to stable coexistence. I also liked that the conditioned media experiments yielded consistent results. It is interesting that the metabolomics suggests that phylogeny and resource utilization are not correlated in these bacteria, and metabolite production correlates with how well a strain grows in a community. I also found it very interesting that the exo-metabolomics did not allow one to predict the positive and negative interactions based on nutrient cross-feeding; this represents an important future challenge for the field.

Questions

117: There is no OD to CFU map. This sounds like an easy and relevant thing to include. How do we know that the experiments are actually starting at 1:1 or 19:1 ratio?

128: How do we know that unimodal distributions indicate no modification in fitness? Doesn't it matter where the peak is?

161: I am confused with the legend of figure S6. As the penalization grows, shouldn't the number of parameters that are less than .01 increase? Shouldn't the red line go up instead of down?

165: I would like to see a table with the estimated parameters in the supplementary, apart from fig S8.

166: Isn't $1e-5$ extremely small?

232: Is this paragraph in contradiction with the previous one? Does this mean that the trajectories are well predicted by the model, yet the final steady states are not? Or was this the model predictions only using the monoculture growth?

279: I don't understand what 'prolonged duration of time' means. Is this just a long relaxation time to an equilibrium?

292: 'Frequency of environmental shifts'? Do the authors mean the dilution rate? Is Fig S15 supposed to show us that increasing dilution rate slows down equilibration?

346: This conditioned media experiments won't reveal any higher order interactions. Perhaps it would be informative to make an experiment where all the spent media from 12 bugs is pooled together and used to make the pairwise competitions? If the trajectory of a pair changes, it might be due to the production of a metabolite by another metabolite in the media.

449: This paragraph is a little mysterious and I am not sure that I am convinced or that I know exactly what the authors are trying to say.

Minor Points

Line 21: 'is a dense of collection'

Line 51: Cooperation and competition are not necessarily the determinant factors of a community.

Line 110: Is the 24hr transfer really a reflection of the gut dynamics or is it mainly to allow the communities to stabilize.

Line 170: In Fig. S7b it seems that there are more negative than positive interactions, which is contrary to what the text says. Maybe some label is incorrect?

Line 176: I am incapable of understanding the labels on the x-axis of fig. S7c.

Line 290: I don't see a difference in figure S15 between the pairs mentioned in text and the other pairs.

Line 329: What is a good and bad coefficient of variation?

Line 388: 'did not the recapitulate the'

Line 634: 'collected at an initial'?
 Line 697: How does the linearization work?
 Line 1026: A verb might make this sentence more clear.

1st Revision - authors' response

16 April 2018

Reviewer #1:

This is an important study that studies interactions between gut microbes using assembled in-vitro communities. This approach obviously has its limits in that it is not in-vivo but it allows detailed study and modelling of defined communities and the inferences of interactions and other ecological properties. This is sorely lacking from the great majority of microbiome work, which often rests upon single time point analysis of in vivo communities. Just for methods alone then, this paper is a major step forward and in addition the authors present a number of interesting and important results that suggest that their approach is the right one. Key amongst these is the finding that pairwise interactions allow one to predict much of the interaction networks of larger communities. There are a lot of negative options out there on these approaches that rest upon the challenges that come with the potential higher order interactions in communities. These opinions often miss the fact that LV like models can certainly deal with this conditionality although it does make everything harder to predict. Moreover, the authors here show evidence that the biology is also on our side in finding that these higher order effects are not dominant in these systems. While this is only in vitro at this point, this is a very promising and important step for the future of predicting these vital and complex communities. I have only a couple of other comments:

1. How reliable is OD as a measure of abundance? Cell adhesion, size and shape can all affect OD. Did the authors plate out cells for CFU and check it matches reasonably well? These are culturable bugs so it is certainly possible.

Every experimental measurement has intrinsic biases including OD600, genome extraction, multi-template PCR and CFU counting. CFU counting is influenced by several factors (PMID 9219225, 28600831): (1) cell adhesion (e.g. chaining, aggregation, biofilm formation) will lead to plating of cell clumps that form a single colony, which will underestimate the number of cells in the culture (Goldman, Emanuel, and Lorrence H. Green, eds. *Practical handbook of microbiology*. CRC Press, 2015), (2) dormant sub-populations (e.g. persistence or lag-phase) that do not form colonies at the time of plating but can switch to an active growing state, will also lead to underestimation of the number of cells in the liquid culture, (3) plating cells on solid agar provides a different selection pressure than liquid media and may underestimate the number of cells in the liquid culture. Previous work has shown that $\text{CFU ml}^{-1} \text{OD}^{-1}$ varies a function of growth stage for *E. coli* (PMID 21829590). At each time point, the different species in the communities can be in distinct growth stages due to a broad range of monospecies lag phases and growth rates (**Appendix Fig. S2**). Several species in the synthetic community display cell adhesion traits including chaining or aggregation, which influences both OD600 and CFU counting. CFU counting is a low throughput method to estimate the total number of viable cells in a culture. As such, carrying out CFU counting measurements for each time point across all sub-communities would have been very challenging.

Since it was not feasible to measure CFU ml^{-1} at each time point for each community using our current methodology (approximately 2247 measurements including biological replicates), we would have needed to use a single estimate of $\text{CFU ml}^{-1} \text{OD}^{-1}$ for each organism. Since the single estimate of $\text{CFU ml}^{-1} \text{OD}^{-1}$ for each organism varies significantly as a function of growth stage, we measured total biomass using OD600 measurements. Our model was trained on absolute abundance estimated using OD600 values and used to predict absolute abundance estimated from OD600 measurements. Therefore, all biases due to OD600 are automatically incorporated into the model training and prediction procedure. Nevertheless, as the reviewers suggested, we carried out measurements of $\text{CFU ml}^{-1} \text{OD}^{-1}$ for all species at early stationary phase (**Fig. R1**). DP was not included in this set due to inconsistent growth on agar plates. Our results show that all organisms have a $\text{CFU ml}^{-1} \text{OD}^{-1}$ ranging between 10^7 - 10^9 . FP measured at 28 hr (late exponential) and 40 hr (early stationary phase) exhibited a 7.3-fold variation in $\text{CFU ml}^{-1} \text{OD}^{-1}$ values ranging between 5.3×10^6 to 3.9×10^7 $\text{CFU ml}^{-1} \text{OD}^{-1}$, illustrating that this metric varies as a function of growth stage. We have now included a

few sentences in our results section about the potential biases in OD600 and CFU counting (lines 117-121).

Figure R1. Box plot of colony forming units (CFU) per ml per OD600 for 11 monospecies. Cultures were inoculated from 75 μ L single-use 25% glycerol stocks into 5 mL of ABB media (Hi-Media) in quadruplicate. ABB media was supplemented with 33 mM acetate (Sigma) for *E. rectale*. Strains in the slow growth and fast growth categories (see Materials and Methods) were incubated at 37°C without shaking for 16 hrs or 41 hrs, respectively. Cultures were mixed 15 times with 1 mL volumes before OD600 was measured on a UV-Vis Spectrophotometer (Shimadzu). The cultures were serially diluted two-fold into ABB media using a Freedom Evo liquid-handling robot (TECAN). 100 μ L of the dilution estimated to produce \sim 100 CFU was plated on 100 x 15 mm ABB agar plates using 4.5 mm glass plating beads (Zymo) and incubated at 37°C without shaking. ABB agar plates were supplemented with 33 mM acetate (Sigma) for *E. rectale*. *F. prausnitzii* was inoculated from 150 μ L single-use 25% glycerol stocks into 10 mL of ABB media (Hi-Media) in quadruplicate and incubated at 37°C without shaking. Following 28 and 40 hours, cultures were mixed 15 times with 1 mL volumes before removing a 2 mL aliquot to measure OD600 and cells were plated as described above. On each box, the red line represents the median, the edges of the box are the 25th and 75th percentiles, the whiskers extend to the most extreme data points. Four biological replicates of each species were analyzed.

2. *What is the basis for the decision of the media used to grow the communities? Given the importance of media for growth and interactions, I think some discussion of this would be helpful.*

We used a rich media referred to as “anaerobic basal broth” (ABB) to grow the monospecies and communities. This media contains peptone, yeast extract, sodium chloride, starch, dextrose, sodium pyruvate, arginine, sodium succinate, L-cysteine, sodium bicarbonate, ferric pyrophosphate, haemin, vitamin K, sodium thioglycollate, and dithiothreitol (pH 6.8). The media was selected to support the growth of all monospecies in the community and was not designed to recapitulate the gut microbiome environment. Nevertheless, we show that the relative abundance of 7 species in the synthetic community is correlated with the median abundance of these species in metagenomic sequencing of human gut samples (**Appendix Fig. S11**). We have now included a sentence about media selection to our results section (lines 105-106).

The goal of our study was to investigate design principles of microbial community assembly and to develop predictive models of multi-species community dynamics in a single well-controlled environment. While our media may not accurately represent the gut microbiome environment, our work provides a foundation for understanding the ecological forces shaping the assembly of anaerobic gut communities. Experiments to investigate the generalizability of our results in different medias that mirror the gut microbiome environment are currently underway.

3. *The authors choose to end by contrasting their results to our Science paper that argued that cooperation can be destabilising for microbial communities. While I realise that this has a nice rhetorical ring for a closing, I find this contrast a bit of a straw man as we never said that cooperation is not important for communities. Indeed, we go to some length discuss the importance of cooperation for things like productivity, which is a major ecological factor, and it is clear that if one species props up another, positive (one or two way) interactions can be important for diversity in a stable system. What we argued, however, is that too much cooperation can be a bad thing owing to the effects on stability, which many non-network-theory people found surprising. Looking at the authors' network in Figure 2 I am struck by the fact that a) the majority of interactions are negative and b) the majority of positive interactions appear to be exploitative (+/-) or commensal*

(+/-) not cooperative (+/+) as implied by the closing paragraph. (It would be good to have numbers on these interaction frequencies if they are not somewhere that I missed). This kind of network is exactly the kind that we predicted would display stability as there will be negative feedbacks in there. One does not need the system to have all negative interactions, one just needs a certain number of these feedback. Indeed, we show (admittedly in our supplement Figure S2) that adding exploitative interactions to a cooperative network can stabilise it, just like adding competitive interactions.

Thank you, Dr. Foster for raising this point. We agree that the balance of positive and negative interactions may promote ecological stability. We have revised the closing paragraph to state (lines 588-600): "Positive interactions were observed frequently in the synthetic human gut microbiome community in contrast to other synthetic communities¹⁷. For example, EL had 6 positive outgoing interactions and displayed the largest impact on community assembly and productivity. These results show that positive interactions played a major role in shaping the dynamics and metabolic efficiency of the synthetic human gut community. Negative interactions have been shown to stabilize cooperative networks by introducing negative feedbacks⁶³. Indeed, networks coupled by negative and positive interactions (+/-) contained 81% of the inferred positive interactions as opposed to mutualism (+/+, 9.5%) or unidirectional positive (+/0, 9.5%) topologies. The prevalent negative interactions in the inferred interaction network may promote ecological stability, whereas the frequent positive interactions modulate community assembly and metabolic efficiencies. Future work will elucidate the plasticity of the ecological network in response to changeable environments and the generalizability of these principles in higher-dimensional microbiomes that mirror the complexity of the natural system."

Reviewer #2:

The manuscript entitled "Deciphering microbial interactions in synthetic human gut microbiome communities" by Venturelli & Carr et al. describes a set of experimental and computational explorations of community assembly using a set of 12 bacterial species representing the diversity in the human gut microbiome. While there are many findings in the paper, the main finding put forward in the abstract is the importance of pairwise interaction in community formation.

This work is intriguing. The results of pairwise competitions and the synthetic community experiments, including dropouts, provide a nice resource for the microbiome community. The ability of the model to predict community formation so well from pairwise growth rates is exciting. While some works have used pairwise interactions to predict the composition of more complex communities (e.g. Friedman 2017), that simple models including only pairwise interactions perform well for human-associated microbes (e.g Fig 3d) is exciting.

This paper makes many claims throughout. A number of claims are not well supported by the data, require further clarification or refinement of methods (e.g. analysis of robustness to outliers), or are contradicted by other parts of the paper. In particular, the stability of the communities on a whole to dropouts (Fig 3a) appears to contradict the importance of pairwise interactions highlighted throughout, and this requires further explanation. The connections between different sections are not always clear, and the methods are not always well explained.

Major comments

The single-species dropouts revealed different organisms to be important to community assembly than suggested by the network hub analysis -- many species with high outgoing interactions have minimal effect in the dropout experiments. Can the authors elaborate further on this? At face value, this does not appear to be consistent with the claims that pairwise interactions are essential to explain most of the data.

The network in **Fig. 2c** represents the sign, magnitude and direction of the inferred inter-species interaction coefficients for the gLV model trained on T3 and the number of incoming and outgoing edges in the network for each species is shown in **Appendix Fig. S7a**. The gLV model has a defined mathematical relationship of species concentrations, growth rates, intra-species interactions and inter-species interactions. Therefore, the contribution of each species to community assembly in the model depends on all model parameters and cannot be predicted by only by the inter-species

interaction coefficients. For example, BO and BV have similar number of outgoing interactions (**Appendix Fig. S7a**) but exhibit significantly different contributions to community assembly (**Fig. 3b**) and abundances in the full community (**Fig. 3e**).

We analyzed the contributions of each organism to community assembly based on the helpful suggestion by the reviewer below. Our results now show that communities lacking EL, BO, BU, CH, BT or DP exhibit differences in community assembly compared to the full community (**Fig. 3b**). Communities lacking BH, ER, PC, CA, FP or BV do not show significant differences in community assembly. Difference in community assembly was computed as $\sum_{i=1}^{11} \frac{1}{7} \sum_{j=1}^7 (\hat{v}_{ji,FULL} - v_{ji,X})^2$ where i and j represent species and time points, respectively. X represents a single-species dropout community lacking the organism X. \hat{v} denotes the renormalized relative abundance of the shared set of species in the full (12-member) and single-species dropout community. We also show that the species relative abundance in the full community at 72 hr plus the sum of the outgoing gLV inter-species interaction coefficients is correlated to the difference in community dynamics (EL is a major driver of this signal). We have clarified this point in the results section by stating (lines 187-191): “The contribution of each species to full community assembly in the model is dictated by a defined mathematical relationship of the monospecies growth rate, intra-species interaction, outgoing and incoming inter-species interactions. Therefore, a prediction about the role of each organism in community assembly requires simulation and analysis of the gLV model.”

The metric used to identify the effect of dropouts is most sensitive to changes in the abundance of the most abundant species. Perhaps an alternative metric could better capture, other, more subtle changes. A simple thing that could be done is to create a 'predicted' community for each dropout by subtracting the removed strain from the final full community composition, and renormalizing. Evenness could be compared to the predicted evenness, Pearson correlations could be calculated, or something else.

We thank the reviewer for the helpful suggestion. We performed this analysis and now include new figures showing the differences in community dynamics (**Fig. 3b**) and Pearson correlation (**Appendix Fig. S12a**). These data show that EL, BO and BU have the largest impact on community structure and is thus consistent with our conclusions.

Interaction terms in these gLV models can be negative because of simple competition for resources (this is why self-self interaction is always negative). This should be stated plainly somewhere for the uninitiated reader, both in the results and in the discussion (e.g. around Line 463).

We have stated that negative interactions can arise due to resource competition in the introduction (lines 56-61) and results (lines 181-183) sections.

FP is a very slow grower in these conditions (Fig S2). Perhaps its discovery as a recipient species is dependent upon the particular conditions of the media? This may be wind up being biological (similar nutrients may be limiting in vivo), but the results may be entirely dependent on the media. Therefore, the claim that FP is ecologically responsive Line 176-178 is overstated (adding "under these conditions" in line could work).

The inferred ecologically couplings are conditional based on environmental context. We have now mentioned that our discovery of FP as a recipient organism may be dependent on the environmental conditions (lines 198-199).

Similarly, the mentioned absence of "context-dependent interactions" in the abstract is not supported, as pairwise interactions may be conditional to this media.

In the abstract, “context-dependent” was referring to higher-order interactions. To clarify this point, we have replaced “context-dependent” by “higher-order interactions” in the abstract (lines 6-7).

Further, a lot of the community in Figure 3 is still not explained by the pairwise data, even in this media. Line 262, "conditional interactions played a minor role in driving community assembly", is also an overstatement. (Probably other locations as well)

The model trained on T3 and T4 explained 72% and 81% of the variance on average of the dynamics of multi-species communities, respectively (**Fig. 3c**). Therefore, these data indicate that the pairwise gLV model was able to explain the majority of the temporal changes in community structures and thus higher-order interactions could only have a moderate contribution to community assembly in these conditions. To clarify this point, we modified this sentence to state (lines 309-312): “In sum, the pairwise gLV model could accurately predict the temporal changes in the majority of species in the multi-species communities, which suggests that higher-order interactions played a minor role in driving community assembly.”

It would be helpful to see lines in Figure 3d showing the predictions from the model without T2 and T3. This would make it easier to evaluate if the inclusion of the pairwise interactions gets much closer to reality than growth rates alone, and why.

We now show model predictions based on training on T1-T4 in the revised **Fig. 3e**.

Line 193-7: How much more predictive power does including the species interactions give than the growth parameters for comparison to metagenomic data?

Appendix Fig. S11 shows the scatter plot of the median relative abundance in metagenomic data (PMID 20203603) vs. the mean relative abundance at 72 hr in the full synthetic community for seven species present in the metagenomics dataset based on our experimental results. Prediction by models trained on T1 (training set: monospecies dataset only) and T4 (training sets: monospecies dataset, PW1, PW2 and full community) did not exhibit a high correlation with the median species relative abundance in the metagenomics dataset (**Fig. R2a,b**). However, the model trained on T4 exhibited a high correlation with the relative abundance in the metagenomics dataset when the outlier BO was removed (**Fig. R2d**). The model trained on T1 excluding BO was not correlated, demonstrating that pairwise inter-species interactions for six species in the synthetic community significantly improved the predictive capability of the model compared to training set T1 (**Fig. R2c**).

Figure R2. Scatter plots of median relative abundance in metagenomic sequencing data vs. predicted relative abundance in the full community for the model trained on (a), (c) T1 or (b), (d) T4. ρ and P denote the Pearson correlation coefficient and p-value, respectively.

Line 212-216: Does this analysis hold if EL is removed? It looks to be driving most of the signal in Figure S12.

Removing EL in **Appendix Fig. S13** significantly reduced the Pearson correlation coefficient ($\rho = 0.13$, $P = 0.71$), demonstrating that EL was a driver of this correlation. We have noted this in the text by stating (lines 260-266): “The species impact score, defined as the species relative abundance in the full community at 72 hr plus the sum of the outgoing gLV inter-species interaction coefficients, was correlated to the difference in community dynamics for single-species dropouts ($\rho = 0.67$, $P < 0.05$, **Appendix Fig. S13**) and EL was the major driver of this correlation. These data suggest that the inferred ecological network and relative abundance pattern could explain the contribution of EL to multi-species community assembly.”

Line 314-316: How more likely are +/- pairs than -/- pairs given the distribution of positive and negative interactions in the observed data set?

The +/- pairs are observed (26%) more frequently than predicted (19.5%), given the distribution of interaction types. The -/- pairs are observed less frequently (33%) than predicted (37%).

Line 314-316: How robust are the described disparities between the topologies to different sets of growth rates?

We analyzed coexistence behavior across a broad range of simulated growth rates for all of the inferred positive/negative and negative/negative interaction networks and included a new

supplementary figure based on this analysis (**Appendix Fig. S18**). These results show that the positive/negative interaction networks are more significantly more robust to variations in growth rates compared to the bidirectional negative interaction networks.

Modeling constraint/ MCMC modeling: I found this section weak, though I am not an expert in MCMC models. A) In the methods, the authors state that “If the chains have converged to the target posterior distribution the PRSF should be close to 1” and then “Our results showed that 81% of parameters had a PRSF less than 3.” Three is not that close to one, so I can’t understand why this is used as evidence of convergence.

To determine if the number of iterations is limiting for convergence, we computed the fraction of parameters that have a PSRF less than 1.5 as a function of the number of iterations (**Fig. R3a**). Our results show that the fraction of parameters with a PSRF less than 1.5 saturates around 500000 iterations. Therefore, longer simulations of the Markov chain are unlikely to significantly increase the number of converged parameters. Our data shows that the magnitude of each parameter and the PSRF are negatively correlated (**Fig. R3b**, $R_s = -0.73$, $P = 0$ where R_s and P represent the Spearman correlation and P-value, respectively), demonstrating that parameters that exhibit poor convergence rates have small magnitudes. Therefore, parameters that have been forced to zero by regularization show poor convergence rates, suggesting that lack of convergence is isolated to a specific set of parameters that do not significantly influence model behaviors.

The PSRF values are correlated to the mean Pearson correlation coefficient between parameter pairs ($R_s = -0.64$, $P = 0$) for the single Markov chain simulated from the inferred parameter set based on T3 (**Fig. R3c**). Even though a fraction of the parameters may not have converged to the stationary distribution, the Metropolis-Hastings MCMC analysis provided information about the correlations between parameters and constraint of the model by our data (**Appendix Fig. S20**). We have updated the Methods section to include information about convergence (lines 808-821). Due to the large number of parameters (156 total), states and model nonlinearities, convergence of the Metropolis Hastings MCMC is difficult. However, we performed a regularized optimization procedure to determine the value of the parameters that balances the goodness of fit with the sparsity of the solution and then evaluated parameter constraints based on Metropolis-Hastings MCMC initialized from this best parameter estimate. Indeed, for all five Markov chains, the mean of each parameter across 570000 iterations was high correlated to the parameter set based on T3 ($\rho \geq 0.99$), demonstrating that MCMC was exploring a single mode of the Posterior distribution.

Figure R3. (a) Fraction of parameters with a potential scale reduction factor (PSRF) less than 1.5 as a function of the number of iterations calculated using four Markov chains that were initialized using randomly sampled starting points for 670000 iterations. The mean of the initial point for each Markov chain was randomly sampled with a mean equal to the inferred parameter set T3 and a standard deviation equal to 0.3. (b) Scatter plot of the magnitude of each parameter based on T3 vs. PSRF. PSRF was computed using four Markov chains that were initialized from randomly sampled starting points with a burn-in period of 100000. R_s and P denote the Spearman correlation coefficient and p-value, respectively. (c) Scatter plot of the mean Pearson correlation vs. PSRF for a single Markov chain simulated from the inferred parameter set based on T3. PSRF was computed using four Markov chains that were initialized from randomly sampled starting points with a burn-in period of 100000. R_s and P denote the Spearman correlation coefficient and p-value, respectively.

The distribution across parameters (histogram) might also be useful.

We simulated four independent chains for 500000 iterations from randomly sampled parameter values using a normal distribution with the mean equal to the parameter estimate based on training set T3 and standard deviations equal to 0.2 (C1), 0.3 (C2) or 0.4 (C3). A burn-in period of 100000 was used for this analysis. The PSRF distributions show that 52%, 51% and 42% of values were less than 1.5 for C1, C2 and C3, respectively (Fig. R4a,b,c). These results show that a higher fraction of parameters converged for the initial parameter vector sampled more closely to the parameter estimate trained on T3. In the paper, we simulated the Markov chain for a total of 670000 iterations and computed the PSRF using a standard deviation of 0.3. Our results show that the 60% of parameters exhibit a PSRF less than 1.5 (Fig. R4d). Due to timing limitations for submission of the

revised manuscript, C1 and C4 were simulated for a total of 500000 iterations. However, we do not expect that simulating the Markov chains for additional iterations would alter our conclusions about the trend shown in **Fig. R4a,b,c**.

Figure R4. Histograms of the potential scale reduction factor (PSRF) of four Markov chains sampled from an initial value that was randomly sampled with a mean equal to the inferred parameter set based on training set T3 and standard deviation equal to (a) 0.2, (b) 0.3 or (c) 0.4 for 500000 total iterations. A burn-in period of 100000 was used to exclude the initial set of samples. The dashed line represents the threshold 1.5 (values closer to 1 indicate convergence to the stationary distribution). The number at the top right corner represents the percent of parameters that had a PSRF less than 1.5. (d) Histogram of PSRF computed using four Markov chains that were initialized using a randomly sampled parameter set with a mean equal to the inferred parameter set based on T3 and standard deviation equal to 0.3. The Markov chains were simulated for 670000 iterations with a burn-in period of 100000 iterations.

B) Line 329-330: How meaningful is this given that the priors were set with a CV of .3?

We assume that the reviewer is referring to the standard deviation of the randomly sampled initial parameter vector for evaluating convergence of the Markov chains. We have now varied the standard deviations of the randomly sampled parameter sets for evaluating convergence (**Fig. R4**). Our results show that the fraction of parameters that converge based on the potential scale reduction factor decrease as a function of the standard deviation.

C) Was the MCMC run in such a way that the sign of an interaction could change? How would this affect the results?

We do not have constraints against sign changes. However, none of the parameters change sign across all iterations of the Markov chain simulated from the parameter estimate based on T3 across 570000 iterations (burn-in period of 100000).

D) Sup Fig 18f: The connection between coefficient magnitude and constraint would be easier to evaluate if a scatter plot of average correlation vs. average absolute effect size were shown.

We have plotted a scatter plot of the magnitude of each parameter vs. the median absolute value of the Pearson correlation coefficient (**Appendix Fig. S20f**). The Pearson correlation is equal to -0.47 and is statistically significant ($P = 9.3e-10$), indicating parameters with a smaller magnitude exhibit a larger median absolute value of the Pearson correlation.

E) Overall, the methods section is very sparse for this section, and it isn't always clear what is being talked about in the text. This is important, as the average reader of MSB does not have expert-level knowledge in these methods. For example, it is not clear in the text that the CV estimates are on the last 200,000 iterations of the chain or if the Pearson correlations are across or within MCMC chains.

We have revised the methods section to add details about the Metropolis-Hastings MCMC method (lines 798-821).

The legend of 18f refers to 'at least 25' Pearson correlations, but the Methods state that 4 MCMC chains were run.

Appendix Fig. S20f is based on a single Markov chain initialized using the parameter estimate based on T3 with a burn-in period of 100,000 iterations. In the previous version of the manuscript, we had compared the magnitude of parameters that displayed a Pearson correlation larger than 0.6 or less than -0.6 with at least 25 other parameters. Therefore, these parameters were highly correlated to at least 25 other parameters in the model. The four Markov chains initialized using randomly sampled starting points were not used for this analysis. We have now replaced this figure with a scatter plot of the parameter magnitude vs. median absolute value of the Pearson correlation coefficient (**Appendix Fig. S20f**).

It is also not clear if the last sentence on page 11 is referring to some step in this MCMC constraint estimation or something else.

We have clarified that the **Appendix Fig. S20f** was based on a single Markov Chain initialized using the inferred parameter set based on training set T3 (lines 401-405).

How much are metabolite secretion measurements affected by randomly lysed cells (e.g. during stationary phase), or cell lysis during sample preparation? Can you rule out that such artifacts are impacting the results?

We thank the reviewer for raising an important point. We measured the exo-metabolomic profiles of monospecies at exponential or early stationary phase (**Appendix Fig. S22**). While we do not expect significant cell lysis at these growth stages, we cannot rule out the possibility that cell lysis impacted our metabolomics data. We have now stated this limitation in our results section (lines 442-446).

How does secretion correlate with growth rate? While secretion could be operationally defined to include cell lysis, this would change the interpretations made in the discussion (lines 496-502).

The total change in metabolite secretion and growth rate exhibit a significant relationship. We have included this figure in the supplement (**Appendix Fig. S23e**) and clarified that secretion is defined to operationally include cell lysis (lines 445-6).

The metabolite analysis is interesting but not connected much to the observed species interactions. Are the interactions predicted by metabolites reflected in the model and/or the spent media experiments? Why or why not?

The metabolite profiling of single species provided insight into the resource utilization and secretion capabilities of monospecies. These data predicted a set of metabolites that could form the molecular basis of microbial interactions (**Appendix Fig. S24**). BU was a major producer of secreted metabolites (**Appendix Fig. S23**) and exhibited a significant impact on community assembly (**Fig. 3b**). However, the overall environmental impact based on metabolite secretion/utilization by individual species did not predict the role of each species in community assembly in some cases. For example, EL had the largest contribution to community assembly and utilized and secreted a moderate number of metabolites (**Figs. 3b, 4a,b, Appendix Fig. S23**). By contrast, CH was a metabolite hub by utilizing and secreting a large number of metabolites and had a moderate influence on community assembly (**Figs. 3b, 4a,b, Appendix Fig. S23**). The total log₂ fold change in metabolite secretion was correlated to the number of negative interactions, suggesting that secreted metabolites may inhibit growth. Our results and discussion sections discuss the relationships between the metabolite profiles, inferred interactions and contributions to full community assembly. There are several potential reasons why the metabolite profiles are not predictive of the inferred interactions and species impact on community assembly: (1) we failed to measure important metabolites mediating inter-species interactions, (2) metabolites that decrease may not be limiting for growth in the community context, (3) secreted metabolites may not be limiting for growth in a community context, or (4) species may exhibit different metabolite utilization or secretion capabilities in a community context compared to monospecies growth. However, investigating these questions is beyond the scope of this study.

How does the analysis in Supplementary Figure 21a change if the outlier is removed?

The conclusion about the lack of correlation between the total log₂ fold change in utilized metabolites vs. total biomass is not altered if the outlier CH is removed from **Appendix Fig. S23a** ($R_s = 0.07$, $P = 0.84$).

Specific comments:

Growth rate of some slow growers (e.g. FP; CA) does not appear to be fast enough to maintain saturation in 1:20 dilutions of single cultures. How was carrying capacity then defined? The authors might refer to the number used for normalization as "max OD".

We now refer to carrying capacity as maximum OD₆₀₀ (line 132).

Line 184: This statement is quite strong and should be supported with some sort of quantitative argument or toned down. It is likely that the phylogenetic signal is quite important but changes quite rapidly, such that only bacteria related below a certain taxonomic level would have similar interaction patterns - and sufficient resolution was provided only within the genus level.

We have revised the text to include more details about the relationship between phylogeny and microbial interactions (lines 216-222): "Hierarchical clustering of the gLV interaction coefficients showed that *Bacteroides* and *Actinobacteria* exhibited similar patterns in outgoing and incoming microbial interactions (**Appendix Fig. S9a,b**). However, the *Firmicutes* clustering pattern did not recapitulate the phylogenetic relationships. Together, these data show that distantly related species can display similar microbial interactions (e.g. BH and DP or PC and CA) and closely related species can exhibit distinct interaction patterns (e.g. BH and ER) (**Fig. 1a**). Therefore, evolutionary similarity was not a global predictor of patterns in the gLV interaction coefficients^{36,37}."

Error bars are needed in Figure S11.

Error bars and replicates are shown in **Appendix Fig. S12** and **Fig. 3b**.

Line 193-7: Why were only 7 species used for comparison to metagenomic data? Not enough information is given here.

The metagenomic dataset contained only 7 of the 12 species in the synthetic community (PMID 20203603). We have clarified this point in the manuscript.

Line 274: Does "history-dependent dynamic behavior in the model" mean that the data used for

determining history-dependence were from the model predictions, not from experiments?

We experimentally observed history-dependent behavior in a set of 8 pairwise communities (**Fig. S5a,b**). To understand the origin of history-dependence and dependence of this behavior on model parameters, we analyzed the inferred pairwise communities that experimentally exhibited history-dependent behavior in our model. We have now clarified this point in our manuscript (lines 229-237).

Line 318-321: The link between the network topology at this level and the observed stability of the gut microbiome is highly speculative. What justification is there to average interaction coefficients across members of a phylum? Why at this taxonomic level and not others? What about the other topologies in this network?

We agree with the reviewer that the justification for analyzing the phylum level was not provided. We now include an order-level network since each order includes at least two species with the exception of DP in *Desulfovibrionales*. The order-level network provides insight into the relationships between groups of species based on evolutionary relatedness. We have revised the text to state (lines 368-374): “We analyzed the network interactions at a higher taxonomic level to illuminate interaction patterns based on evolutionary relatedness. The inferred gLV interaction coefficients based on T3 were averaged across species associated with a given order since this level encompassed at least two species with the exception of DP in *Desulfovibrionales*. Several orders were connected by positive and negative interactions in the order-level interaction network, illustrating that the positive and negative interaction motif was consistent at higher taxonomic rankings (**Appendix Fig. S19**).”

Line 23-5: This is highly speculative language. Colonization resistance is an emergent property, not quite a core function. Chemical transformations by microbes have some effects on the human host, but it is not clear that any currently known reaction is a "core function" of the microbiome.

We have revised this sentence to state (lines 25-28): “Functions of the gut microbiota are partitioned among genetically distinct populations that interact to perform complex chemical transformations and exhibit emergent properties such as colonization resistance at the community-level.”

Line 41-2: The cited paper did not show stability on the scale of decades.

We have revised the sentence to state (lines 35-37): “Constituent strains of the gut microbiota have been shown to persist in an individual over long periods of time, demonstrating that the gut microbiota exhibits stability over time¹⁴.”

*Line 48-9: F/B ratio has not stood the test of time (<http://mbio.asm.org/content/7/4/e01018-16.full>)
Line 50-1: This statement is strong and very speculative given the state of the field.*

We agree with the reviewer that there is conflicting evidence that the F/B ratio provides information about human health and disease. As such, we have revised this paragraph (lines 34-43).

Which strains were used for each organism?

We have listed the strains (ATCC or DSM number) in the methods section (lines 612-617).

Typos and polish: A version of Figure S8 and S5 clustered by phylogeny would be helpful for interpreting claims of lack of phylogenetic clustering of interactions.

We have revised **Appendix Fig. S5a,b** by clustering based on phylogenetic relatedness. We have also included **Appendix Fig. S9b** that shows the inferred gLV inter-species interaction coefficients based on T3 sorted by phylogenetic relatedness.

Fig S2: Data points and "shading" are not distinguishable in PDF

We have revised **Appendix Fig. S2** to show error bars and model fits based on training sets T1-T4.

Line 170: Amenalism is jargon. While this word is defined in the following parentheses, it is not obvious in to a non- specialist reader what this notation means.

We have revised this sentence to state (lines 185-7): Pairwise networks were enriched for unidirectional negative (-/0, 36%), bidirectional negative (-/-, 32%) and positive and negative (+/-, 26%) species couplings (**Appendix Fig. S7b**).”

Line 232: Clarify what "the model" means here.

We revised the text to state (lines 225-227): “To validate the gLV model based on training set T3, time-resolved measurements of relative abundance of the full (12-member) and all single-species dropout communities (11-member consortia, **Fig. 3a**) and total community biomass (**Appendix Fig. S10a**) were performed.”

Line 552: What is a sequencing plate here?

Sequencing plate referred to a deep-well plate containing the arrayed microbial communities. We have replaced “sequencing plate” with “deep-well plate” (lines 625 and 629).

Line 553: Breath -> Breathe

We have corrected this error in the manuscript.

Fig 3c is not necessary for a main figure.

We have moved the heat-map showing the Pearson correlation coefficient across all time points and multi-species communities to **Appendix Fig. S14b**.

Figure 4c would be strengthened with a schematic graphic similar to that shown in 4b.

We revised **Fig. 4c** to show the network topology.

Line 314-316: I get this sentence, but it was extremely hard to parse.

We have revised this sentence to state (lines 362-364): “Pairs of species that exhibited coexistence were linked by positive/negative and mutual inhibition in 50% and 25% of cases, illustrating that the positive/negative interaction motif was frequently associated with coexistence behavior (**Appendix Fig. S5b**).”

Line 324-5: Please qualify this sentence to explain that you are talking about microbiome modeling papers. Do you mean that no microbiome modeling manuscripts have evaluated uncertainty, or just some?

We have revised this sentence to state (lines 379-381): “Pairs of species that exhibited coexistence were linked by positive/negative and mutual inhibition in 50% and 25% of cases, illustrating that the positive/negative interaction motif was frequently associated with coexistence behavior (**Appendix Fig. S5b**).”

Bipartate is jargon.

We removed “bipartite” from the results section.

Descriptive text on the panels in Fig 5 would help orient readers

We have added sub-titles to **Fig. 5a,b,d,and e**.

Legend for Fig 3d should state that model T3 is used here.

We modified the figure legend to state which training sets are displayed in **Fig. 3e**.

Parameter symbols aren't neatly introduced until the Methods, but are used in the main figures. Perhaps remove symbolic representation from main figures or introduce in main text.

We have now defined the model and parameters in the main text (lines 162-165).

Fig S3: A line at $x=1$ would be helpful for interpreting this Figure.

We now show the $x = 1$ line in **Appendix Fig. S3a**.

Reviewer #3:

Despite the known health importance of the human gut microbiome, surprisingly little is known about how individual species within the gut interact and the consequences of these interactions for the structure of the community. As a first step to address these issues, the authors used a collection of 12 gut bacteria to make pairwise competition experiments, while tracking the relative abundances across time with multiplexed 16s sequencing, and the total biomass with light absorbance. Three data sets were constructed: the first one includes only the single species growing; the second one includes also the pairwise competition experiments starting at 1:1 ratio; and the third one adds up the competition experiments starting at 95-5% and 5-95%. These temporal datasets were used to estimate the parameters of a Lotka- Volterra model; the authors find that the pairwise interactions do a pretty good of predicting the outcome of a variety of multispecies competitions (better than the monoculture growth alone). This conclusion reinforces what previous recent studies have said about communities in other environments, but it is worth stressing the importance of studying gut microbes despite the extra difficulty of working anaerobically. Interestingly, the metabolomics provided surprisingly little insight into the interactions, which I view as a significant warning or challenge to the field. I think that this paper is a nice contribution and could be published with only minor changes.

The authors measure a number of positive interactions between species, and this is contrasted with the results of Foster & Bell (2012). However, it is important to note that Foster & Bell only measured total biomass (not individual fractions), meaning that a positive interaction in that paper actually requires a very strong (and likely two-way) positive interaction. If the authors analyzed their data in a similar way would they see any examples of an increase in total biomass? This is relevant because in our past work we saw many examples of one-way facilitation but no examples of two-way mutualisms in which the total biomass was increased (Friedman et al, 2017).

We have included a new supplementary figure (**Appendix Fig. S8**) showing a scatter plot of predicted productivity based on the sum of monospecies growth response (null model) and measured productivity for pairwise communities. Productivity is defined as the integral of the growth response based on OD600 over a 23.5 hr period. The predicted productivity null model is computed by summing the monospecies integrals of OD600 over a 23.5 hr period. Community productivity was reduced compared to the null model for 38% of communities, consistent with the prevalence of negative interactions in the inferred network. Pairwise communities BT, BV; BT, BU; BT, BO; BV, BU; BV, BO and BU, BO exhibited significantly lower productivity compared to the null model. Consistent with this result, the inferred gLV interaction network demonstrated two-way negative interactions for these six consortia that exhibited significantly lower measured productivities compared to the null models. Pairwise communities EL, BH; EL, BU; EL, BT; EL, BO, EL, BV and CH, ER exhibited significantly larger productivity compared to the null model predictions. Four of these consortia were coupled by a unidirectional positive interaction (EL, BT; EL, BU; EL, BO; EL, BV, CH, ER) and EL, BH had bidirectional positive interactions. Together, these results show that community productivity was increased for specific networks that have both one and two-way positive interactions. We have included a description of these results in our manuscript (lines 202-215).

More generally, I don't understand how the authors could observe so many positive interactions if a majority of the competitions lead to competitive exclusion.

The majority of inferred inter-species interactions based on T3 are negative (56% negative and 21%

positive). We have updated **Appendix Fig. S5b** to show the signs of the inferred inter-species interactions overlaid on a heat-map that displays the classification of pairwise behaviors based on experimental data. In all communities that displayed competitive exclusion, the inferred interactions are unidirectional negative (-/0), bidirectional negative (-/-) or positive/negative (+/-).

*We find that the species that gain from other species are typically those that grow very poorly; was this the case for *F. prausnitzii*, which had five incoming positive interactions?*

Yes, FP has a low growth rate and maximum OD600 (**Appendix Fig. S2**) and the fitness of FP was significantly enhanced by specific members of the community.

As far as I understand, the sequencing of 16S was done every 12 hours, whereas the dilutions were done every 24 hours. Do the authors see examples of apparent "oscillations" in relative frequencies? We normally only make frequency measurements with a period that is equal to (or a multiple of) the dilution period because otherwise there can be oscillations that are in some sense an artifact (the enlarged time trace in Fig 2a may have such dynamics...).

We thank the reviewer for raising this question. We do see potential "oscillations" in relative abundance in a subset of pairwise communities (e.g. BV, FP in PW1; BV, BO in PW1; BH, EL in PW2; CH, ER in PW2; CA, BH in PW1; CH, CA in PW1 and CH, DP in PW2). We selected an interval of 24 hr for serial transfer intervals to prevent strains with long lag phases (e.g. PC, CA, FP in **Appendix Fig. S2**) from being eliminated from communities at the serial dilution stage. A 12 hr interval was selected to provide information about the community dynamics prior to stationary phase. Potential "oscillations" in relative abundance due to our choice of serial transfer and sampling times do not alter our model or conclusions. However, the reviewer's question is intriguing and we are currently investigating the effects of transfer time on community structure as part of a separate effort in the lab. We added a justification about the selection for a 12 hr sampling and 24 hr serial transfer time in our results section (lines 106-110).

Also, in Fig 1b the time trace coloring and the fraction bar plot coloring seem to imply something that is not true. The fractions in the bar plot are within a given well (that has a given color - either blue or green), whereas the bar plot has both colors shown.

We thank the reviewer for catching this error. We have updated **Fig. 1b** with an accurate schematic of our experimental design.

I found the discussion of history-dependent (HD) pairs to be a bit confusing. Ideally, the authors could distinguish between slow relaxation to an equilibrium and bistability. I understand that this is not always possible, but I couldn't quite figure out what the authors think is happening with their pairs. We see some pairs that are probably slow relaxation, other pairs that show clear bistability (with only one species surviving at equilibrium), other pairs that show bistability with one of the states corresponding to coexistence (which is not an allowed outcome of LV), and a few ugly pairs that are difficult to categorize. As a reader, I would have appreciated a better sense of how the authors history-dependent pairs break down among these categories.

We have clarified in the manuscript that history-dependent (HD) behaviors are defined as slow relaxation to a monostable equilibrium (lines 314-348). We show which communities exhibit HD behavior based on experimental data in **Appendix Fig. S5b**. The final time point (72 hr) may not represent steady-state for the communities. In fact, HD behavior illustrates that communities can exhibit slow relaxation to equilibrium and may not reach steady-state by the final measured time point. Therefore, the final time point may not represent a steady-state solution to the pairwise gLV model. However, it is also possible that the gLV model cannot accurately represent the dynamics of all pairwise communities.

We used a quantitative threshold applied to the data to classify communities into *coexistence*, *competitive exclusion (dominance)*, *HD* or *other* categories. Communities in the "other" category did not quantitatively satisfy the criteria for the coexistence, HD and dominance categories and many of these communities displayed weak HD behavior. We analyzed the differences in the inferred models of the communities that experimentally exhibited HD behavior (**Fig. 4, Appendix S16**). According to our model trained on T3, only one community BU, BT exhibited bistability. The

remaining 7 HD communities displayed a slow relaxation to monostable equilibrium.

I have particular concerns about the categorization that the authors are using, which involves parameterization of the LV model. Instead, I would much prefer it if the authors directly look at the data and tell us what outcome they think is happening (a LV parameterization of an outcome not allowed by LV model will likely be misleading). I would have appreciated a separate figure on this topic where the authors show the raw data and with a description of what the authors think is happening.

The classification of pairwise community behaviors is based on experimental data and not the model (**Appendix Fig. S5b**). The classification is based on the raw data in **Appendix Fig. S5a**. Here is a description about how the data was classified into community behaviors in **Appendix Fig. S5b**: *History-dependence* required at least 40% variation in relative abundance at 72 hr for each organism based on two conditions inoculated using distinct initial species proportions. *Dominance* occurred when one organism was at least 95% of the community at 72 hr and the community did not exhibit a history-dependent response. *Coexistence* required that both species were at least 5% of the community at 72 hr and the consortium did not exhibit a history-dependent response. Communities in the *other* category did not quantitatively satisfy the criteria for *history-dependence*, *dominance* or *coexistence*.

I like the idea that the pairs with +/- interactions establish a negative feedback loop that leads to stable coexistence. I also liked that the conditioned media experiments yielded consistent results. It is interesting that the metabolomics suggests that phylogeny and resource utilization are not correlated in these bacteria, and metabolite production correlates with how well a strain grows in a community. I also found it very interesting that the exo-metabolomics did not allow one to predict the positive and negative interactions based on nutrient cross-feeding; this represents an important future challenge for the field.

Questions

117: There is no OD to CFU map. This sounds like an easy and relevant thing to include. How do we know that the experiments are actually starting at 1:1 or 19:1 ratio?

Every experimental measurement has intrinsic biases including OD600, genome extraction, multi-template PCR and CFU counting. CFU counting is influenced by several factors (PMID 9219225, 28600831): (1) cell adhesion (e.g. chaining, aggregation, biofilm formation) will lead to plating of cell clumps that form a single colony, which will underestimate the number of cells in the culture (Goldman, Emanuel, and Lorrence H. Green, eds. *Practical handbook of microbiology*. CRC Press, 2015), (2) dormant sub-populations (e.g. persistence or lag-phase) that do not form colonies at the time of plating but can switch to an active growing state, will also lead to underestimation of the number of cells in the liquid culture, (3) plating cells on solid agar provides a different selection pressure than liquid media and may underestimate the number of cells in the liquid culture. Previous work has shown that $\text{CFU ml}^{-1} \text{OD}^{-1}$ varies a function of growth stage for *E. coli* (PMID 21829590). At each time point, the different species in the communities can be in distinct growth stages due to a broad range of monospecies lag phases and growth rates (**Fig. S2**). Several species in the synthetic community display cell adhesion traits including chaining or aggregation, which influences both OD600 and CFU counting. CFU counting is a low throughput method to estimate the total number of viable cells in a culture. As such, carrying out CFU counting measurements for each time point across all sub-communities would have been very challenging.

Since it was not feasible to measure CFU ml^{-1} at each time point for each community using our current methodology (approximately 2247 measurements including biological replicates), we would have needed to use a single estimate of $\text{CFU ml}^{-1} \text{OD}^{-1}$ for each organism. Since the single estimate of $\text{CFU ml}^{-1} \text{OD}^{-1}$ for each organism varies significantly as a function of growth stage, we measured total biomass using OD600 measurements. Our model was trained on absolute abundance estimated using OD600 values and used to predict absolute abundance estimated from OD600 measurements. Therefore, all biases due to OD600 are automatically incorporated into the model training and prediction procedure. Nevertheless, as the reviewers suggested, we carried out measurements of $\text{CFU ml}^{-1} \text{OD}^{-1}$ for all species at early stationary phase (**Fig. R1**). DP was not included in this set due to inconsistent growth on agar plates. Our results show that all organisms have a $\text{CFU ml}^{-1} \text{OD}^{-1}$

ranging between 10^7 - 10^9 . FP measured at 28 hr (late exponential) and 40 hr (early stationary phase) exhibited a 7.3-fold variation in $\text{CFU ml}^{-1} \text{OD}^{-1}$ values ranging between 5.3×10^6 to 3.9×10^7 $\text{CFU ml}^{-1} \text{OD}^{-1}$, illustrating that this metric varies as a function of growth stage. We have now included a few sentences in our results section about the potential biases in OD600 and CFU counting (lines 117-121). Starting the experiments with exactly 1:1 or 19:1 ratio based on cell number is not critical for our model-guided approach to infer microbial interactions.

Figure R1. Box plot of colony forming units (CFU) per ml per OD600 for 11 monospecies. Cultures were inoculated from 75 μL single-use 25% glycerol stocks into 5 mL of ABB media (Hi-Media) in quadruplicate. ABB media was supplemented with 33 mM acetate (Sigma) for *E. rectale*. Strains in the slow growth and fast growth categories (see Materials and Methods) were incubated at 37°C without shaking for 16 hrs or 41 hrs, respectively. Cultures were mixed 15 times with 1 mL volumes before OD600 was measured on a UV-Vis Spectrophotometer (Shimatzu). The cultures were serially diluted two-fold into ABB media using a Freedom Evo liquid-handling robot (TECAN). 100 μL of the dilution estimated to produce ~ 100 CFU was plated on 100 x 15 mm ABB agar plates using 4.5 mm glass plating beads (Zymo) and incubated at 37°C without shaking. ABB agar plates were supplemented with 33 mM acetate (Sigma) for *E. rectale*. *F. prausnitzii* was inoculated from 150 μL single-use 25% glycerol stocks into 10 mL of ABB media (Hi-Media) in quadruplicate and incubated at 37°C without shaking. Following 28 and 40 hours, cultures were mixed 15 times with 1 mL volumes before removing a 2 mL aliquot to measure OD600 and cells were plated as described above. On each box, the red line represents the median, the edges of the box are the 25th and 75th percentiles, the whiskers extend to the most extreme data points. Four biological replicates of each species were analyzed.

128: How do we know that unimodal distributions indicate no modification in fitness? Doesn't it matter where the peak is?

The reviewer is correct that the peak of the distribution is important. We have now included an additional panel (**Appendix Fig. S3b**) that shows the coefficient of variation (CV) of the distributions in **Appendix Fig. S3a** and revised the text accordingly. Our results show that *Bacteroides* and CH have the lowest coefficient of variation, indicating that the fitness of these species were not significantly altered in the presence of a second organism. The fitness levels of species that displayed bimodal (DP, FP, BH, CA, PC), long-tail (ER, EL) distributions and/or high CV values were modified in the presence of a second organism (lines 132-138).

161: I am confused with the legend of figure S6. As the penalization grows, shouldn't the number of parameters that are less than .01 increase? Shouldn't the red line go up instead of down?

The number of parameters with an absolute value greater than 0.01 decreases as a function of the regularization coefficient λ . The legend in the previous version of the manuscript had an error and has now been corrected.

165: I would like to see a table with the estimated parameters in the supplementary, apart from fig S8.

We have added tables of inferred parameters based on T1-T4 as **Appendix Figures S25-28**.

166: Isn't $1e-5$ extremely small?

The impact of the inter-species interaction coefficient depends on the other parameters in the model

including the growth rate and intra-species interaction coefficient. We have clarified this point in our results section (lines 175-180). In our model, Interaction coefficients with a magnitude less than $1e-3$ are not expected to change the steady-state species abundance based on the inferred growth rate and intra-species interaction values. In addition, the number of parameters with an absolute value above the threshold does not change significantly between $1e-6$ and $1e-2$ (**Fig. R5**). Indeed, only six negative inter-species interaction coefficients that are within this range including $\alpha_{ER,CA}$ ($-2.5e-6$), $\alpha_{BO,PC}$ ($-1e-4$), $\alpha_{BV,PC}$ ($-8.7e-6$), $\alpha_{ER,EL}$ ($-3e-5$), $\alpha_{EL,DP}$ ($-2.9e-4$), $\alpha_{ER,DP}$ ($-1.2e-6$). Therefore, variations in the threshold within this range do not significantly alter our conclusions. We have included a sentence in our results section about the relationship between significant threshold and network connectivity. Finally, we only applied a threshold for visualizing the network in **Fig. 2c** and included all parameters for model training and prediction.

Figure R5. Number of parameters with absolute value larger than a threshold for the model trained on T3.

232: *Is this paragraph in contradiction with the previous one? Does this mean that the trajectories are well predicted by the model, yet the final steady states are not? Or was this the model predictions only using the monoculture growth?*

The sentence does not contradict the previous paragraph. As the reviewer noted, this sentence required clarification. In **Appendix Fig. S14a**, we show a scatter plot of the predicted OD600 value of monospecies vs. the measured relative abundance in the full community at 72 hr. We

have revised this sentence to state (lines 281-284): “The predicted monospecies OD600 based on the model trained on T3 and the fraction of each species in the full community at 72 hr were not correlated ($\rho = 0.46$, $P = 0.13$), corroborating that monospecies growth failed to forecast the structure of the full community (**Appendix Fig. S14a**).”

279: *I don't understand what 'prolonged duration of time' means. Is this just a long relaxation time to an equilibrium?*

We are referring to a slow relaxation to equilibrium. We have revised these paragraphs for clarity (lines 314-348). Our results show that communities can display a broad range of times to converge to a steady-state composition and this timescale depends on the nature and strength of the inter-species interactions.

292: *'Frequency of environmental shifts'? Do the authors mean the dilution rate? Is Fig S15 supposed to show us that increasing dilution rate slows down equilibration?*

Appendix Fig. S16 shows that the time required for the system to reach equilibrium increases as a function of the dilution rate. We have revised the text to state this conclusion (lines 337-348).

346: *This conditioned media experiments won't reveal any higher order interactions. Perhaps it would be informative to make an experiment where all the spent media from 12 bugs is pooled together and used to make the pairwise competitions? If the trajectory of a pair changes, it might be due to the production of a metabolite by another metabolite in the media.*

Using the pairwise gLV model, we show that up to 81% of the variance of all 11-member communities can be explained on average. Therefore, monospecies growth rates and pairwise interactions are major drivers of multi-species community dynamics in these conditions. Spent media is a particular condition that may not accurately represent the dynamic feedbacks that shape microbial interactions and community assembly. Conditioned media has been depleted of key resources and has lower energy availability. As such, spent media can induce couplings that do not exist for community growth in richer media. Therefore, we would need to validate and understand the conditioned media responses to interpret the results and accurately predict inter-species interactions. We now included a paragraph in the discussion section discussing the limitations of conditioned media (lines 561-570). In addition, we added a few sentences about the potential for higher-order interactions in lower dimensional assemblages (e.g. 3-species, lines 501-503). Finally, the proposed experiment to infer pairwise interactions in a mixture of conditioned medias from the 10 remaining community members is beyond the scope of this study.

449: This paragraph is a little mysterious and I am not sure that I am convinced or that I know exactly what the authors are trying to say.

In this paragraph, we are arguing that communities are highly dynamic and community assembly can require variable lengths of time. For example, our model shows that communities with specific interactions and parameters can exhibit history-dependent behavior over periods of days. Microbiomes and microbial communities are continuously perturbed by environmental stimuli, which may shift the operating point of the system (e.g. dietary shifts or exposure to antibiotics). Following these perturbations, it may take a long time for the community to assemble to a steady-state composition. Nevertheless, the steady-state assumption has been used to analyze the human gut microbiome and this assumption may not always be valid due to slow relaxation times to a steady-state community composition (PMID 27279224).

Minor Points

Line 21: 'is a dense of collection'

We revised this sentence to state: “The gut microbiome is a dense collection” (lines 23-4).

Line 51: Cooperation and competition are not necessarily the determinant factors of a community.

We have revised this sentence to state (lines 52-53): “Cooperation and competition generate positive and negative feedbacks in microbial communities and influence ecological functional activities and stability.”

Line 110: Is the 24hr transfer really a reflection of the gut dynamics or is it mainly to allow the communities to stabilize.

We performed serial transfers to monitor community assembly over many generations and allow time for communities to converge to a steady-state composition. We selected a 24 hr period to allow species that exhibit long lag phases to persist in communities based on the distribution of lag phases in **Appendix Fig. S2**. The gut microbiome environment is a dynamically changing environment. Nutrients are replenished via periodic dietary inputs and bacteria are eliminated through colonic transit time. The serial transfers establish a temporally changing environment that mirrors the changeable environment of the human gut microbiome. We have revised the text to clarify these points (lines 106-112).

Line 170: In Fig. S7b it seems that there are more negative than positive interactions, which is contrary to what the text says. Maybe some label is incorrect?

We thank the reviewer for pointing out the error in the text. We have corrected this sentence in the revised manuscript and now state: “Of these interactions, 56% and 21% were negative and positive, respectively (**Fig. 2c**)” (lines 180-181).

Line 176: I am incapable of understanding the labels on the x-axis of fig. S7c.

We have revised the x-axis labels in **Appendix Fig. S7c** to show a “+” or “-“ indicating that the incoming inter-species interaction coefficient for FP $\alpha_{FP,X}$ was set to the inferred value based on training set T3 or zero, respectively.

Line 290: I don't see a difference in figure S15 between the pairs mentioned in text and the other pairs.

In the previous version of our manuscript, we had highlighted three pairwise communities that displayed the most significant history-dependent responses scored by the magnitude of the Euclidean distance across a range of dilution rates. We have revised the text to highlight the five pairwise communities that displayed significant history-dependent behaviors in the presence of inter-species interactions (BO, BT; DP, PC; CA, PC; BO, BV and BO, BU) (lines 341-343).

Line 329: What is a good and bad coefficient of variation?

In the Metropolis Hastings MCMC analysis, the coefficient of variation (CV) is defined as the standard deviation of the Markov chain divided by the mean of the Markov chain. The CV represents the variation in the parameter across all iterations of the Markov chain in relation to the mean. Coefficients of variation less than 1 are considered low-variance whereas greater than 1 indicates high-variance. Our results show that the coefficient of variation was less than 1 for all parameters (**Appendix Fig. S20a**), illustrating that the parameters were constrained by the data.

Line 388: 'did not the recapitulate the'

We have corrected this error in the main text.

Line 634: 'collected at an initial'?

We revised the sentence to state (lines 704-5): “Samples were collected immediately following cell inoculation.”

Line 697: How does the linearization work?

We have described the linearization procedure in the Materials and Methods section (lines 768-770).

Line 1026: A verb might make this sentence more clear.

The line numbers are different in our version of the manuscript and therefore we are not able to identify the specific sentence that the reviewer is describing. We have read through the figure legends and made a few modifications to the text to improve clarity.

2nd Editorial Decision

20 April 2018

Thank you for submitting your revised manuscript. We are now satisfied with the modifications made and I am glad to inform you that your study is now suitable for publication in Molecular Systems Biology.

Before we can formally accept your study for publication, we would ask you to address a few remaining editorial issues listed below:

- We have implemented a "model curation service" for papers that include mathematical models. This is done together with Prof. Jacky Snoep and the FAIRDOM team and it is still in a pilot phase (therefore has not yet been announced officially). In brief, the aim is to enhance reproducibility and add value to papers including mathematical models. Jacky Snoep's summary on the model (*Model Curation Report*) is pasted below. We would ask you to address the issues raised in his report when you submit your revision. I hope you agree that the model curation is a useful initiative and we would of course love to hear your feedback or suggestions. Let me know if you have any questions

regarding this, and if you need help with generating the SBML files we can bring you in contact with Prof. Snoep.

Model Curation Report

It is MSB policy to make mathematical models available in standard formats, so that readers of MSB manuscripts can reproduce the simulations results shown in the manuscript. The model is correctly described in the manuscript, and looks so simple that one could argue that it is not necessary to make model files available to the readers. However, upon a closer look it becomes evident that with a large number of species there are potentially many interactions, making it rather tedious to simulate the model if it needs to be coded by hand.

Therefore, I suggest that the authors make their models available in SBML format. Probably the easiest way to do this, is to make one large model with all species and all interactions included. The user can then decide on a specific scenario by simply adding a non-zero initial concentration for the species that the user wants to analyse. I would recommend to make four models with identical structures available, but with different parameter values according to the four training sets, i.e. T1, T2, T3 or T4.

Using the parameters in the Tables, displayed in the figures S25-S28, I tried to reproduce some of the simulations shown in Figure 2 (two species interactions). I managed to reproduce the simulations qualitatively but not quantitatively. For a quantitative simulation I would need to have the initial conditions for the individual species. In addition, my simulation results showed monotonous curves that were continuous in time, and not the scattered dynamics as shown for the simulation lines in Figure 2.

If the authors do not have a tool for generating SBML files we can provide them with a basic text file that they can extend to include the full model and then upload this to one of our servers to generate SBML.

Corresponding Author Name: Ophelia S Venturelli

Manuscript Number: Molecular Systems Biology

Manuscript Number: MSB-17-8157